# NAS-Bench-301 and the Case for Surrogate Benchmarks for Neural Architecture Search

## Abstract

The most significant barrier to the advancement of Neural Architecture Search (NAS) is its demand for large computational resources, which hinders scientifically sound empirical evaluations. As a remedy, several tabular NAS benchmarks were proposed to simulate runs of NAS methods in seconds. However, all existing tabular NAS benchmarks are limited to extremely small architectural spaces since they rely on exhaustive evaluations of the space. This leads to unrealistic results that do not transfer to larger search spaces. To overcome this fundamental limitation, we propose NAS-Bench-301, the first surrogate NAS benchmark, using a search space containing $10^{18}$ architectures, many orders of magnitude larger than any previous tabular NAS benchmark. After motivating the benefits of a surrogate benchmark over a tabular one, we fit various regression models on our dataset, which consists of ∼60k architecture evaluations, and build surrogates via deep ensembles to also model uncertainty. We benchmark a wide range of NAS algorithms using NAS-Bench-301 and obtain comparable results to the true benchmark at a fraction of the real cost. Finally, we show how NAS-Bench-301 can be used to generate new scientific insights.

## 1 Introduction

Neural Architecture Search (NAS) promises to advance representation learning by automatically finding architectures that facilitate the learning of strong representations for a given dataset. NAS has already achieved state-of-the-art performance on many tasks (Real et al., 2019; Liu et al., 2019a; Saikia et al., 2019; Elsken et al., 2020) and to create resource-aware architectures (Tan et al., 2018; Elsken et al., 2019a; Cai et al., 2020). For a review, we refer to Elsken et al. (2019b).

Despite many advancements in terms of both efficiency and performance, empirical evaluations in NAS are still problematic. Different NAS papers often use different training pipelines, different search spaces and different hyperparameters, do not evaluate other methods under comparable settings, and cannot afford enough runs for testing significance. This practice impedes assertions about the statistical significance of the reported results, recently brought into focus by several authors (Yang et al., 2019; Lindauer & Hutter, 2019; Shu et al., 2020; Yu et al., 2020).

To circumvent these issues and enable scientifically sound evaluations in NAS, several tabular benchmarks (Ying et al., 2019; Zela et al., 2020b; Dong & Yang, 2020; Klyuchnikov et al., 2020) have been proposed recently (see also Appendix A.1 for more details). However, all these benchmarks rely on an exhaustive evaluation of *all* architectures in a search space, which limits them to unrealistically small search spaces (so far containing only between 6k and 423k architectures). This is a far shot from standard spaces used in the NAS literature, which contain more than $10^{18}$ architectures (Zoph & Le, 2017; Liu et al., 2019b). This discrepancy can cause results gained on existing tabular NAS benchmarks to not generalize to realistic search spaces; e.g., promising anytime results of local search on existing tabular NAS benchmarks were shown to not transfer to realistic search spaces (White et al., 2020b). To address these problems, we make the following contributions:

1. We present *NAS-Bench-301*, a surrogate NAS benchmark that is first to cover a realistically-sized search space (namely the cell-based search space of DARTS (Liu et al., 2019b)), containing more than $10^{18}$ possible architectures. This is made possible by estimating their performance via a surrogate model, removing the constraint to exhaustively evaluate the entire search space.

2. We empirically demonstrate that a surrogate fitted on a subset of architectures can in fact model the true performance of architectures *better* than a tabular benchmark (Section 2).

3. We analyze and release the NAS-Bench-301 training dataset consisting of ∼60k fully trained and evaluated architectures, which will also be publicly available in the Open Graph Benchmark (Hu et al., 2020) (Section 3).

4. Using this dataset, we thoroughly evaluate a variety of regression models as surrogate candidates, showing that strong generalization performance is possible even in large spaces (Section 4).

5. We utilize NAS-Bench-301 as a benchmark for running various NAS optimizers and show that the resulting search trajectories closely resemble the ground truth trajectories. This enables sound simulations of thousands of GPU hours in a few seconds on a single CPU machine (Section 5).

6. We demonstrate that NAS-Bench-301 can help in generating new scientific insights by studying a previous hypothesis on the performance of local search in the DARTS search space (Section 6).

To foster reproducibility, we open-source all our code and data in a public repo: `https://anonymous.4open.science/r/3f99ef91-c472-4394-b666-5d464e099aca/`

## 2 MOTIVATION – CAN WE DO BETTER THAN A TABULAR BENCHMARK?

We start by motivating the use of surrogate benchmarks by exposing an issue of tabular benchmarks that has largely gone unnoticed. Tabular benchmarks are built around a costly, exhaustive evaluation of *all* possible architectures in a search space, and when an architecture's performance is queried, the tabular benchmark simply returns the respective table entry. The issue with this process is that the stochasticity of mini-batch training is also reflected in the performance of an architecture $i$, hence making it a random variable $Y_i$. Therefore, the table only contains results of a few draws $y_i \sim Y_i$ (existing NAS benchmarks feature up to 3 runs per architecture). Given the variance in these evaluations, a tabular benchmark acts as a simple estimator that assumes *independent* random variables, and thus estimates the performance of an architecture based only on previous evaluations of the same architecture. From a machine learning perspective, knowing that similar architectures tend to yield similar performance, and that the variance of individual evaluations can be high (both shown to be the case by Ying et al. (2019)), it is natural to assume that better estimators may exist. In the remainder of this section, we empirically verify this hypothesis and show that surrogate benchmarks can provide *better* performance estimates than tabular benchmarks based on *less* data.

**Setup** We choose NAS-Bench-101 (Ying et al., 2019) as a tabular benchmark for our analysis and a Graph Isomorphism Network (GIN, Xu et al. (2019a)) as our surrogate model.[1] Each architecture $x_i$ in NAS-Bench-101 contains 3 validation accuracies $y_i^1, y_i^2, y_i^3$ from training $x_i$ with 3 different seeds. We excluded all diverged models with less than 50% validation accuracy on any of the three evaluations in NAS-Bench-101. We split this dataset to train the GIN surrogate model on one of the seeds, e.g., $\mathcal{D}^{train} = \{(x_i, y_i^1)\}_i$ and evaluate on the other two, e.g., $\mathcal{D}^{test} = \{(x_i, \bar{y}_i^{23})\}_i$, where $\bar{y}_i^{23} = (y_i^2 + y_i^3)/2$.

| Model | Mean Absolute Error (MAE) | | |
|---|---|---|---|
| | 1, [2, 3] | 2, [1, 3] | 3, [1, 2] |
| Tab. | 4.534*e*-3 | 4.546*e*-3 | 4.539*e*-3 |
| Surr. | **3.446e-3** | **3.455e-3** | **3.441e-3** |

Table 1: MAE between performance predicted by a tab./surr. benchmark fitted with one seed each, and the true performance of evaluations with the two other seeds. Test seeds in brackets.

We emphasize that training a surrogate to model a search space is not a typical inductive regression task but rather a transductive one. By definition of the search space, the set of possible architectures is known ahead of time (although it may be very large), hence a surrogate model does not have to generalize to out-of-distribution data if the training data covers the space well.

**Results** We compute the mean absolute error $\text{MAE} = \frac{\sum_i |\hat{y}_i - \bar{y}_i^{23}|}{n}$ of the surrogate model trained on $\mathcal{D}^{train} = \{(x_i, y_i^1)\}_i$, where $\hat{y}_i$ is predicted validation accuracy and $n = |\mathcal{D}^{test}|$. Table 1 shows that the surrogate model yields a lower MAE than the tabular benchmark, i.e. $\text{MAE} = \frac{\sum_i |y_i^1 - \bar{y}_i^{23}|}{n}$. We also report the mean squared error and Kendall tau correlation coefficient in Table 6 in the Appendix showing that the ranking between architectures is also predicted better by the surrogate.

---

[1]We used a GIN implementation by Errica et al. (2020); see Appendix B for details on training the GIN.

We repeat the experiment in a cross-validation fashion w.r.t to the seeds and conclude: *In contrast to a single tabular entry, the surrogate model learns to smooth out the noise.*[2]

Next, we fit the GIN surrogate on subsets of $\mathcal{D}^{train}$ and plot how its performance scales with the amount of training data used in Figure 1. The surrogate model performs better than the tabular benchmark when the training set has more than $\sim$21,500 architectures. Note that $\mathcal{D}^{test}$ remains the same as in the previous experiment, i.e., it includes all architectures in NAS-Bench-101. As a result, we conclude that: *A surrogate model can yield strong predictive performance when only a subset of the search space is available as training data.*

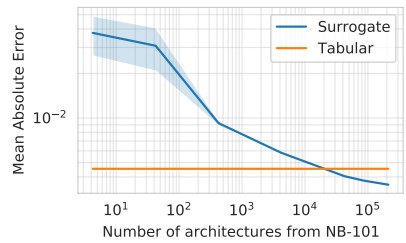

Figure 1: Number of architectures used for training the GIN surrogate model vs MAE on the NAS-Bench-101 dataset.

These empirical findings suggest that we can create reliable surrogate benchmarks for much larger and more realistic NAS spaces, which are infeasible to be exhaustively evaluated as done by tabular benchmarks. In the remainder of the paper, we focus on creating such a benchmark.

## 3 THE NAS-BENCH-301 DATASET

We now describe the *NAS-Bench-301 dataset* which consists of $\sim$60k architectures and their performances on CIFAR-10 (Krizhevsky, 2009) sampled from the most popular NAS cell search space: the one from DARTS (Liu et al., 2019b). We use this dataset not only to fit surrogate models but also to gain new insights, such as which regions of the architecture space are being explored by different NAS methods, or what the characteristics of architectures are that work well.

### 3.1 DATA COLLECTION

Since the DARTS search space (detailed description in Appendix C.1) is far too large to be exhaustively evalu-

| | NAS methods | # eval |
|---|---|---|
| | RS (Bergstra & Bengio, 2012) | 23746 |
| Evolution | DE (Awad et al., 2020) | 7275 |
| | RE (Real et al., 2019) | 4639 |
| BO | TPE (Bergstra et al., 2011) | 6741 |
| | BANANAS (White et al., 2019) | 2243 |
| | COMBO (Oh et al., 2019) | 745 |
| One-Shot | DARTS (Liu et al., 2019b) | 2053 |
| | PC-DARTS (Xu et al., 2020) | 1588 |
| | DrNAS (Chen et al., 2020) | 947 |
| | GDAS (Dong & Yang, 2019) | 234 |

Table 2: NAS methods used to cover the search space.

ated, care has to be taken when sampling the architectures which will be used to train the surrogate models. Sampling should yield a good overall coverage of the architecture space while also providing a special focus on the well-performing regions that optimizers tend to exploit.

Our principal methodology is inspired by Eggensperger et al. (2015), who collected unbiased data about hyperparameter spaces by random search, as well as biased and dense samples in high-performance regions by running hyperparameter optimizers. This is desirable for a surrogate benchmark since we are interested in evaluating NAS methods that exploit such good regions of the space. Table 2 lists the NAS methods we used to collect such samples and the respective number of samples. Additionally, we evaluated $\sim$1k architectures in poorly-performing regions for better coverage and another $\sim$10k for the analysis conducted on the dataset and surrogates. We refer to Appendices C.2 and C.3 for details on the data collection

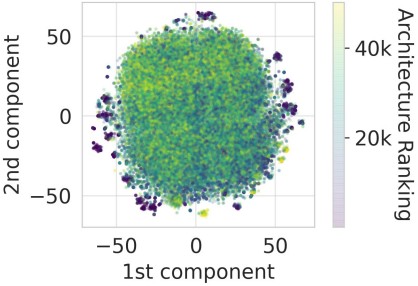

Figure 2: t-SNE visualization of the sampled architectures.

and the optimizers, respectively. We would like to point out that in hindsight adding training data of well-performing regions may be less important for a surrogate NAS benchmark than for a surrogate HPO benchmark, which we demonstrated in Appendix E.3. We argue that this is a result of HPO

---

[2]We do note that the average estimation error of tabular benchmarks could be reduced by a factor of $\sqrt{k}$ by performing $k$ runs for each architecture. The error of a surrogate model would also shrink when the model is based on more data, but as $k$ grows large tabular benchmarks would become competitive with surrogate models.

search spaces containing many configurations which yield disfunctional models, which is less common for architectures in many NAS search spaces, hence allowing random search to give us good coverage of the space.

In Figure 2, we visualize the overall coverage of the search space as well as the similarity between sampled architectures using t-SNE (van der Maaten & Hinton, 2008). Besides showing a good overall coverage, some well-performing architectures in the search space form distinct clusters which are mostly located outside the main cloud of points. This clearly indicates that architectures with similar performance are close to each other in the architecture space. Additionally, we observe that different optimizers sample different types of architectures, see Figure 9 in the Appendix.

### 3.2 Performance statistics

Figure 3 shows the validation error on CIFAR-10 (Krizhevsky, 2009) of all sampled architectures in relation to the model parameters and training runtime. Generally, as expected, models with more parameters are more costly to train but achieve lower validation errors. We also find that different NAS methods yield quite different performance distributions (see Appendix C.4 for their individual performances). Validation and test errors are highly correlated with a Kendall tau rank correlation of $\tau = 0.852$ (Spearman rank corr. 0.969), minimizing the risk of overfitting on the validation error.

Furthermore, we find that cells of all depths can reach a good performance, but shallow topologies are slightly favored in our setting (see Figure 10 in the Appendix). Also, a small number of parameter-free operations (e.g., skip connections) can benefit the performance but featuring many of these significantly deteriorates performance. For the full analysis, see Appendix C.5.

Following standard practice in modern NAS papers (e.g., Liu et al. (2019b)), we employ various data augmentation techniques during training for more reliable estimates of an architecture's performance. For a description of our full training pipeline, please see Appendix C.6.

### 3.3 Noise in Architecture Evaluations

As discussed in Section 2, the noise in architecture evaluations can be large enough for surrogate models to yield more realistic estimates of architecture performance than a tabular benchmark based on a single evaluation per architecture. To study the magnitude of this noise on NAS-Bench-301, we evaluated 500 architectures randomly sampled from our Differential Evolution (DE) (Awad et al., 2020) run with 5 different seeds each.[3] We find a mean standard deviation of $1.6e{-}3$ for the final validation accuracy which is slightly less than the noise observed in NAS-Bench-101 (Ying et al., 2019); one possible reason for this could be a more robust training pipeline. Figure 12 in the Appendix shows that, while the noise tends to be lower for the best architectures, a correct ranking based on a single evaluation is still difficult. Finally, we compare the MAE when estimating the architecture performance from only one sample to the results from Table 1. Here, we also find a slightly lower MAE of $1.38e{-}3$ than for NAS-Bench-101.

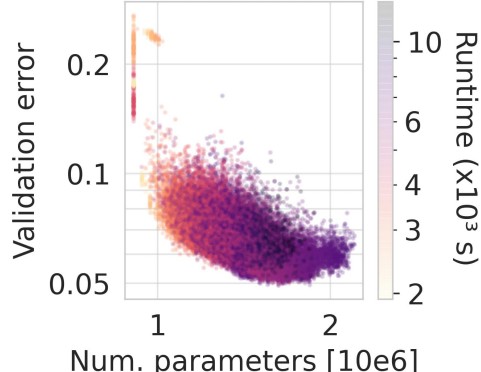

Figure 3: Number of parameters against val. error with model training time as colorbar.

## 4 Fitting Surrogate Models on the NAS-Bench-301 Dataset

We now focus on creating a surrogate model. To that end, we evaluated a wide range of regression models on the NAS-Bench-301 dataset. In principle, any such model can give rise to a surrogate NAS benchmark, but models that fit the true performance better yield surrogate NAS benchmarks whose characteristics are more similar to the ones of the true benchmark. Therefore, we naturally

---

[3] We chose DE because it both explored and exploited well, see Figure 9 in the Appendix.

strive for the best-fitting model. We emphasize that in this work we do not attempt to introduce a new regression model but rather build on the shoulders of the architecture performance prediction community.

## 4.1 SURROGATE MODEL CANDIDATES

Deep Graph Convolutional Neural Networks are frequently used as NAS predictors (Friede et al., 2019; Wen et al., 2019; Ning et al., 2020). In particular, we choose the GIN since several works have found it to perform well on many benchmark datasets (Errica et al., 2020; Hu et al., 2020; Dwivedi et al., 2020). We use the publicly available implementation from the Open Graph Benchmark (Hu et al., 2020) and refer to Appendix D.2 for further details.

We compare the GIN to a variety of common regression models. We evaluate Random Forests (RF) and Support Vector Regression (SVR) using implementations from scikit-learn (Pedregosa et al., 2011). We also compare to the tree-based gradient boosting methods XGBoost (Chen & Guestrin, 2016). LGBoost (Ke et al., 2017) and NGBoost (Duan et al., 2020), recently used for predictor-based NAS (Luo et al., 2020). We comprehensively review architecture performance prediction in Appendix A.2.

## 4.2 EVALUATING THE DATA FIT

Similarly to Wen et al. (2019) and Baker et al. (2017), we assess the quality of the data fit via the coefficient of determination ($R^2$) and the Kendall rank correlation coefficient ($\tau$). Since Kendall $\tau$ is sensitive to noisy evaluations that change the rank of an architecture, we follow the recent work by Yu et al. (2020) and use a sparse Kendall Tau (sKT), which ignores rank changes at 0.1% accuracy precision, by rounding the predicted validation accuracy prior to computing $\tau$.

All hyperparameters of the surrogate models were tuned using BOHB (Falkner et al., 2018) as a black-box optimizer; details on their respective hyperparameter search spaces are given in Table 7 in the appendix. We use train/val/test splits (0.8/0.1/0.1) stratified across the NAS methods used for the data collection. This means that the ratio of architectures from a particular optimizer is constant across the splits, e.g. the test set contains 50% of its architectures from RS since RS was used to obtain 50% of the total architectures we trained and evaluated. We provide additional details on the preprocessing of the architectures for the surrogate models in Appendix D.1. As Table 3 shows, the three best-performing models are LGBoost, XGBoost and GIN; we therefore focus our analysis on these in the following.

| Model | Test | |
|---|---|---|
| | $R^2$ | sKT |
| LGBoost | **0.892** | 0.816 |
| XGBoost | 0.832 | **0.817** |
| GIN | 0.832 | 0.778 |
| NGBoost | 0.810 | 0.759 |
| $\mu$-SVR | 0.709 | 0.677 |
| MLP (Path enc.) | 0.704 | 0.697 |
| RF | 0.679 | 0.683 |
| $\epsilon$-SVR | 0.675 | 0.660 |

Table 3: Performance of different regression models fitted on the NB-301 dataset.

In addition to evaluating the data fit on our data splits, we investigate the impact of parameter-free operations and the cell topology in Appendices D.6 and D.7, respectively. We find that all of LGBoost, XGBoost and GIN accurately predict the drop in performance when increasingly replacing operations with parameter-free operations in a normal cell.

| | Model | No RE | No DE | No COMBO | No TPE | No BANANAS | No DARTS | No PC-DARTS | No DrNAS | No GDAS |
|---|---|---|---|---|---|---|---|---|---|---|
| $R^2$ | LGB | **0.917** | **0.892** | **0.919** | **0.857** | 0.909 | -0.093 | **0.826** | 0.699 | 0.429 |
| | XGB | 0.907 | 0.888 | 0.876 | 0.842 | **0.911** | -0.151 | 0.817 | 0.631 | **0.672** |
| | GIN | 0.856 | 0.864 | 0.775 | 0.789 | 0.881 | **0.115** | 0.661 | **0.790** | 0.572 |
| sKT | LGB | **0.834** | **0.782** | **0.833** | **0.770** | 0.592 | **0.780** | **0.721** | 0.694 | 0.595 |
| | XGB | 0.831 | 0.780 | 0.817 | 0.762 | **0.596** | 0.775 | 0.710 | **0.709** | **0.638** |
| | GIN | 0.798 | 0.757 | 0.737 | 0.718 | 0.567 | 0.765 | 0.645 | 0.706 | 0.607 |

Table 4: Leave One-Optimizer-Out performance of the best surrogate models.

## 4.3 LEAVE ONE-OPTIMIZER-OUT ANALYSIS

Since the aim of NAS-Bench-301 is to allow efficient benchmarking of novel NAS algorithms, it is necessary to ensure that the surrogate model can deliver accurate performance estimation on data from trajectories by unseen NAS methods. Similarly to Eggensperger et al. (2015), we therefore perform a form of cross-validation on the optimizers we used for data collection, i.e. we leave out all data collected by one of the NAS methods entirely during training (using a stratified 0.9/0.1 train/val split over the other NAS methods). Then, we predict the unseen results from the left-out NAS method to evaluate how well the models extrapolate to the region covered by the 'unseen' method. We refer to this as the leave-one-optimizer-out (LOOO) setting.

**Results**  The results in Table 4 show that the rank correlation between the predicted and observed validation accuracy remains high even when a well-performing optimizer such as RE is left out. Predicting BANANAS in the LOOO fashion yields a lower rank correlation, because it focuses on well-performing architectures that are harder to rank; however, the high $R^2$ shows that the fit is still good.

Conversely, leaving out DARTS causes a low $R^2$ but still high sKT; this is due to architectures with many skip connections in the DARTS data that are overpredicted (further discussed in Section 5.2). For full details, Figure 16 in the appendix provides scatter plots of the predicted vs. true performance for each NAS method.

## 4.4 NOISE MODELLING

Ensemble methods are commonly used to improve predictive performance (Dietterich, 2000). Moreover, ensembles of deep neural networks, so-called deep ensembles, have been proposed as a simple way to obtain predictive uncertainty (Lakshminarayanan et al., 2017). We therefore create an ensemble of 10 base learners for each of our three best performing models (GIN, XGB, LGB) using a 10-fold cross-validation for our train and validation split, as well as different initializations. We use the architectures with multiple evaluations (see Section 3.3) to mirror the analysis in the motivation in Section 2. We train using only one evaluation per architecture (i.e., seed 1) and take the mean accuracy of the remaining ones as groundtruth (i.e., seeds 2-5). We then compare against a tabular model with just one evaluation (seed 1).

Table 5 shows that the GIN and LGB surrogate models yield estimates closer to groundtruth than the table lookup based on one evaluation. This confirms our main finding from Section 2, but this time on a much larger search space. We also compare the predictive distribution of our ensembles to the groundtruth. To that end, we assume the noise in the architecture performance to be normally distributed and compute the Kullback–Leibler (KL) divergence between the groundtruth accuracy distribution and

| Model | MAE 1, [2,3,4,5] | Mean $\sigma$ | KL div. |
|---|---|---|---|
| Tabular | 1.38e−3 | undef. | undef. |
| GIN | **1.13e-3** | 0.6e−3 | **16.4** |
| LGB | 1.33e−3 | 0.3e−3 | 68.9 |
| XGB | 1.51e−3 | 0.3e−3 | 134.4 |

Table 5: Metrics for the selected surrogate models on 500 architectures that were evaluated 5 times.

predicted distribution. We find the GIN ensemble to quite clearly provide the best estimate.

To allow evaluations of multi-objective NAS methods, and to allow using "simulated wallclock time" on the x axis of plots, we also predict the runtime of architecture evaluations. For this, we train an LGB model with the runtime as targets (see Appendix D.4 for details). Runtime prediction is less challenging than performance prediction, resulting in an excellent fit of our LGB runtime model on the test set (sKT: 0.936, $R^2$: 0.987). Other metrics of architectures, such as the number of parameters and multiply-adds, do not require a surrogate model but can be queried exactly.

## 5 NAS-BENCH-301 AS A SURROGATE NAS BENCHMARK

Having assessed the ability of the surrogate models to model the search space, we now use NAS-Bench-301 to benchmark various NAS algorithms.

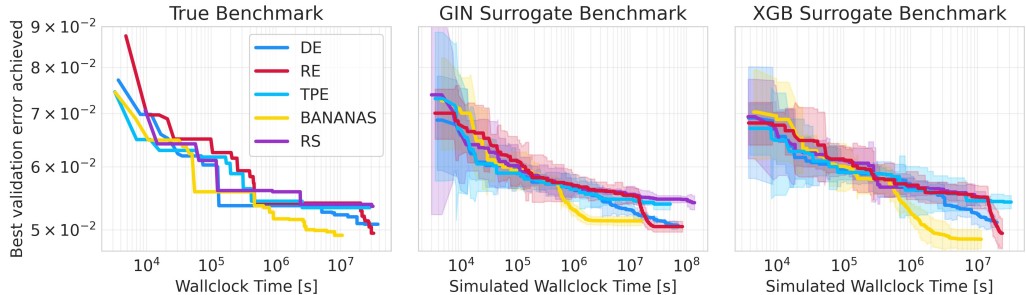

Figure 4: Anytime performance of different optimizers on the real benchmark (left) and the surrogate benchmark (GIN (middle) and XGB (right)) when training ensembles on data collected from all optimizers. Trajectories on the surrogate benchmark are averaged over 5 optimizer runs.

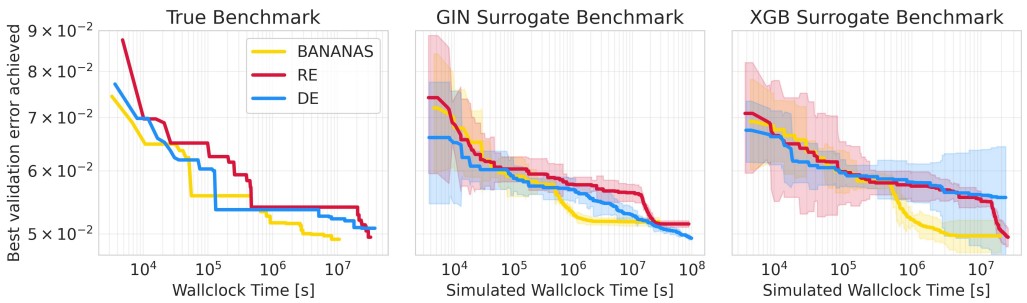

Figure 5: Anytime performance of blackbox optimizers, comparing performance achieved on the real benchmark and on surrogate benchmarks built with GIN and XGB in an LOOO fashion.

## 5.1 BLACKBOX OPTIMIZERS

We first compare the trajectories on the true benchmark and on the surrogate benchmark for blackbox optimizers when training the surrogate on all data. For the true benchmark, we show the trajectories contained in our dataset (based on a single run, since we could not afford repetitions due to the extreme compute requirements of 115 GPU days for a single run). For the evaluations on the surrogate, on the other hand, we can trivially afford to perform multiple runs. For the surrogate trajectories, we use an identical initialization for the optimizers (e.g., initial population for RE) but evaluations of the surrogate benchmark are done by sampling from the surrogate model's predictive distribution for the architecture at hand, leading to different trajectories.

**Results (all data)**  As Figure 4 shows, both the XGB and the GIN surrogate capture behaviors present on the true benchmark. For instance, the strong improvements of BANANAS and RE are also present on the surrogate benchmark at the correct time. In general, the ranking of the optimizers towards convergence is accurately reflected on the surrogate benchmark. Also, the initial random exploration of algorithms like TPE, RE and DE is captured as the large initial variation in performance indicates. Notably, the XGB surrogate ensemble exhibits a high variation in well-performing regions as well and seems to slightly underestimate the error of the best architectures. The GIN surrogate, on the other hand, shows less variance in these regions but slightly overpredicts for the best architectures.

Note, that due to the size of the search space, random search stagnates and cannot identify one of the best architectures even after tens of thousands of evaluations, with BANANAS finding better architectures orders of magnitude faster. This stands in contrast to previous NAS benchmarks. For instance, NAS-Bench-201 (Dong & Yang, 2020) only contains 6466 unique architectures in total, causing the median of random search runs to find the best architecture after only 3233 evaluations.

To simulate benchmarking of novel NAS methods, we expand on the leave-one-optimizer-out analysis (LOOO) from Section 4.3 and assess each optimizer with surrogate benchmarks based on data excluding that gathered by said optimizer. We again compare the trajectories obtained from 5 runs on the surrogate benchmark to the groundtruth.

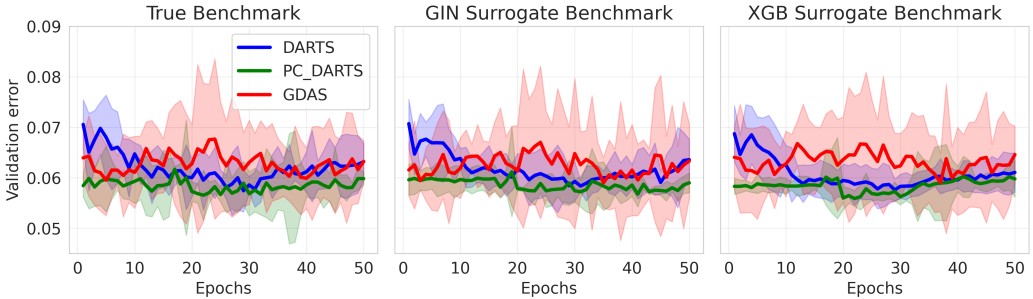

Figure 6: Anytime performance of one-shot optimizers, comparing performance achieved on the real benchmark and on surrogate benchmarks built with GIN and XGB in a LOOO fashion.

**Results (LOOO)**   Figure 5 shows the trajectories in the leave-one-optimizer-out setting. The XGB and GIN surrogates again capture the general behavior of different optimizers well, illustrating that characteristics of new optimization algorithms can be captured with the surrogate benchmark. Leaving out DE appears to be a bigger problem for XGB than GIN, pointing to advantages of the smooth embedding learned by the GIN compared to gradient-boosting.

In an additional experiment, we found that surrogates built on only well-performing architectures (92% and above) yielded poor extrapolation to worse architectures, but that surrogate benchmarks based on them still yielded realistic trajectories. We attribute this to NAS optimizers' focus on good architectures. For details, see Appendix E.2. We also investigate whether it is possible to create benchmarks only on random architectures in Appendix E.3, and find that we can indeed obtain realistic trajectories but lose some predictive performance in the well-performing regions. Nevertheless, such benchmarks have the advantage of not possibly favouring any NAS optimizer used for the generation of training data, and we thus recommend their release in addition to the benchmarks based on the full training data.

### 5.2   One-shot Optimizers

NAS-Bench-301 can also be used to monitor the behavior of one-shot NAS optimizers throughout their search phase, by querying the surrogate model with the currently most promising discrete architecture. This can be extremely useful in many scenarios since uncorrelated proxy and true objectives can lead to potential failure modes, e.g., to a case where the found architectures contain only skip connections in the normal cell (Zela et al., 2020a;b; Dong & Yang, 2020) (we study such a failure case in Appendix E.1 to ensure robustness of the surrogates in said case). We demonstrate this use case in a similar LOOO analysis as for the black-box optimizers, using evaluations of the discrete architectures from each search epoch of multiple runs of DARTS, PC-DARTS and GDAS as ground-truth. Figure 6 shows that the surrogate trajectories closely resemble the true trajectories.

## 6   Using NAS-Bench-301 to Drive NAS Research

We finally use our new benchmark to perform a case study that demonstrates how NAS-Bench-301 can drive NAS research. Coming up with research hypotheses and drawing conclusions when prototyping or evaluating NAS algorithms on less realistic benchmarks is difficult, particularly when these evaluations require high computational budgets. NAS-Bench-301 alleviates this dilemma via its cheap and reliable estimates.

To showcase such a scenario, we evaluate Local Search[4] (LS) on our surrogate benchmark and the actual DARTS benchmark. White et al. (2020b) concluded that LS does not perform well on such a large space by running it for 11.8 GPU days ($\approx 10^6$ seconds), and we are able to reproduce the same results via NAS-Bench-301 in a few seconds (see Fig 7). While White et al. (2020b) could not afford longer runs (nor repeats), on NAS-Bench-301 this is trivial. Doing so suggests that LS shows qualitatively different behavior when run for an order of magnitude longer, transitioning from being

---

[4]We use the implementation and settings of Local Search provided by White et al. (2020b).

the worst method to being one of the best. We verified this suggestion by running LS for longer on the actual DARTS benchmark (also see Fig 7). This allows us to revise the initial conclusion of White et al. (2020b) to: *LS is also state-of-the-art for the DARTS search space, but only when given enough time.*

This case study shows how NAS-Bench-301 was already used to cheaply obtain hints on a research hypothesis that lead to correcting a previous finding that only held for short runtimes. We look forward to additional uses along such lines.

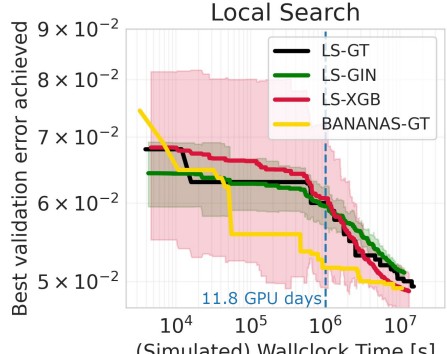

Figure 7: Case study results for Local Search. GT is the ground truth, GIN and XGB are results on NAS-Bench-301.

## 7 CONCLUSIONS & GUIDELINES FOR USING NAS-BENCH-301

We proposed NAS-Bench-301, the first surrogate NAS benchmark and first to cover a realistic search space which is orders of magnitude larger than all previous tabular NAS benchmarks. After motivating the benefits of a surrogate benchmark over a tabular one, we described the strategy used to collect the data which we used to fit our selected surrogate models and evaluated their predictive performance. Lastly, we demonstrated that our surrogate benchmark can accurately simulate real anytime performance trajectories of various NAS methods at a fraction of the true cost and can lead to new scientific findings. We hope that NAS-Bench-301 will allow the NAS practitioner to quickly prototype and benchmark NAS algorithms on the currently most used search space, without requiring large computational resources. We also argue that NAS-Bench-301 could also be used to monitor one-shot optimizers during their search phase, to detect failure cases early on. Finally, the ideas and methods discussed in our work trivially transfer to other search spaces or datasets, allowing for the design of many interesting surrogate benchmarks in the future.

Finally, we want to mention the risk that prior knowledge about the surrogate model in NAS-Bench-301 could lead to the design of algorithms that may overfit to the surrogate benchmark. To this end, we recommend the following best practices to ensure a safe and fair benchmarking of NAS methods on NAS-Bench-301 and future surrogate benchmarks:

- The surrogate model should be treated as a black-box function, hence only be used for performance prediction and not exploited to extract, e.g., gradient information.
- We discourage benchmarking methods that internally use the same model as the surrogate model picked in NAS-Bench-301 (e.g. GNN-based Bayesian optimization should not only be benchmarked using the GIN surrogate benchmark).
- We encourage running experiments on versions of NAS-Bench-301 (and other, future NAS surrogate benchmarks) that are based on (1) all available training architectures and (2) only architectures collected with uninformed methods, such as random search or space-filling designs. As shown in Appendix E.3, (1) yields better predictive models, but (2) avoids any potential bias (in the sense of making more accurate predictions for architectures explored by a particular type of NAS optimizer) and can still yield strong benchmarks.
- In order to ensure comparability of results in different published papers, we ask users to state the benchmark's version number. We will continuously collect more training data and further improve the surrogate model predictions. So far, we release NB301-XGB-v1.0, NB301-GIN-v1.0, NB301-XGB-rand-v1.0, and NB301-GIN-rand-v1.0.

Due to the flexibility of surrogate NAS benchmarks to cover arbitrary search spaces, we expect NAS-Bench-301 to be the first of many such benchmarks. We collect best practices for the creation of new surrogate benchmarks in Appendix F. Having access to a variety of benchmarks is essential to the development and evaluation of new NAS methods. We therefore encourage the community to expand the scope of current NAS benchmarks to different search spaces, datasets, and problem domains utilizing surrogate benchmarks to cover large spaces.

**Acknowledgements** We thank the anonymous reviewers for suggesting very insightful experiments, in particular the experiments for NAS benchmarks based only on random architectures.

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

## A  RELATED WORK

### A.1  EXISTING NAS BENCHMARKS

Benchmarks for NAS were introduced only recently with NAS-Bench-101 (Ying et al., 2019) as the first among them. NAS-Bench-101 is a tabular benchmark consisting of ∼423k unique architectures in a cell structured search space evaluated on CIFAR-10 (Krizhevsky, 2009). To restrict the number of architectures in the search space, the number of nodes and edges was given an upper bound and only three operations are considered. One result of this limitation is that One-Shot NAS methods can only be applied to subspaces of NAS-Bench-101 as demonstrated in NAS-Bench-1Shot1 (Zela et al., 2020b).

NAS-Bench-201 (Dong & Yang, 2020), in contrast, uses a search space with a fixed number of nodes and edges, hence allowing for a straight-forward application of one-shot NAS methods. However, this limits the total number of unique architectures to as few as 6466. NAS-Bench-201 includes evaluations of all these architectures on three different datasets, namely CIFAR-10, CIFAR-100 (Krizhevsky, 2009) and Downsampled Imagenet 16×16 (Chrabaszcz et al., 2017), allowing for transfer learning experiments.

NAS-Bench-NLP (Klyuchnikov et al., 2020) was recently proposed as a tabular benchmark for NAS in the Natural Language Processing domain. The search space resembles NAS-Bench-101 as it limits the number of edges and nodes to constrain the search space size resulting in 14k evaluated architectures.

### A.2  NEURAL NETWORK PERFORMANCE PREDICTION

In the past, several works have attempted to predict the performance of neural networks by extrapolating learning curves (Domhan et al., 2015; Klein et al., 2017; Baker et al., 2017). A more recent line of work in performance prediction focuses more on feature encoding of neural architectures. Peephole (Deng et al., 2017) and TAPAS (Istrate et al., 2019) both use an LSTM to aggregate information about the operations in chain-structured architectures. On the other hand, BANANAS (White et al., 2019) introduces a path-based encoding of cells that automatically resolves the computational equivalence of architectures.

Graph Neural Networks (GNNs) (Gori et al., 2005; Kipf & Welling, 2017; Zhou et al., 2018; Wu et al., 2019) with their capability of learning representations of graph-structured data appear to be a natural choice to learning embeddings of NN architectures. Shi et al. (2019) and Wen et al. (2019) trained a Graph Convolutional Network (GCN) on a subset of NAS-Bench-101 (Ying et al., 2019) showing its effectiveness in predicting the performance of unseen architectures. Moreover, Friede et al. (2019) propose a new variational-sequential graph autoencoder (VS-GAE) which utilizes a GNN encoder-decoder model in the space of architectures and generates valid graphs in the learned latent space.

Several recent works further adapt the GNN message passing to embed architecture bias via extra weights to simulate the operations such as in GATES (Ning et al., 2020) or integrate additional information on the operations (e.g. flop count) (Xu et al., 2019b). Tang et al. (2020) chose to operate GNNs on relation graphs based on architecture embeddings in a metric learning setting, allowing to pose NAS performance prediction as a semi-supervised setting.

## B  TRAINING DETAILS FOR THE GIN IN THE MOTIVATION

We set the GIN to have a hidden dimension of 64 with 4 hidden layers resulting in around ∼40k parameters. We trained for 30 epochs with a batch size of 128. We chose the MSE loss function and add a logarithmic transformation to emphasize the data fit on well-performing architectures.

| Model | Mean Squared Error (MSE) | | | Kendall tau | | |
|---|---|---|---|---|---|---|
| | 1, [2, 3] | 2, [1, 3] | 3, [1, 2] | 1, [2, 3] | 2, [1, 3] | 3, [1, 2] |
| Tab. | 5.44$e$-5 | 5.43$e$-5 | 5.34$e$-5 | 0.83 | 0.83 | 0.83 |
| Surr. | **3.02e-5** | **3.07e-5** | **3.02e-5** | **0.87** | **0.87** | **0.87** |

Table 6: MSE and Kendall tau correlation between performance predicted by a tab./surr. benchmark fitted with one seed each, and the true performance of evaluations with the two other seeds (see Section 2). Test seeds in brackets.

# C  NAS-BENCH-301 DATASET

## C.1  SEARCH SPACE

We use the same architecture search space as in DARTS (Liu et al., 2019b). Specifically, the normal and reduction cell each consist of a DAG with 2 input nodes (receiving the output feature maps from the previous and previous-previous cell), 4 intermediate nodes (each adding element-wise feature maps from two previous nodes in the cell) and 1 output node (concatenating the outputs of all intermediate nodes). Input and intermediate nodes are connected by directed edges representing one of the following operations: Sep. conv $3 \times 3$, Sep. conv $5 \times 5$, Dil. conv $3 \times 3$, Dil. conv $5 \times 5$, Max pooling $3 \times 3$, Avg. pooling $3 \times 3$, Skip connection.

## C.2  DATA COLLECTION

To achieve good global coverage, we use random search to evaluate $\sim$23k architectures. We note that space-filling designs such as quasi-random sequences, e.g. Sobol sequences (Sobol', 1967), or Latin Hypercubes (McKay et al., 2000) and Adaptive Submodularity (Golovin & Krause, 2011) may also provide good initial coverage.

Random search is supplemented by data which we collect from running a variety of optimizers, representing Bayesian Optimization (BO), evolutionary algorithms and One-Shot Optimizers. We used Tree-of-Parzen-Estimators (TPE) (Bergstra et al., 2011) as implemented by Falkner et al. (2018) as a baseline BO method. Since several recent works have proposed to apply BO over combinatorial spaces (Oh et al., 2019; Baptista & Poloczek, 2018) we also used COMBO (Oh et al., 2019). We included BANANAS (White et al., 2019) as our third BO method, which uses a neural network with a path-based encoding as a surrogate model and hence scales better with the number of function evaluations. As two representatives of evolutionary approaches to NAS, we chose Regularized Evolution (RE) (Real et al., 2019) as it is still one of the state-of-the art methods in discrete NAS and Differential Evolution (Price et al., 2006) as implemented by Awad et al. (2020). Accounting for the surge in interest in One-Shot NAS, our collected data collection also entails evaluation of architectures from search trajectories of DARTS (Liu et al., 2019b), GDAS (Dong & Yang, 2019), DrNAS (Chen et al., 2020) and PC-DARTS (Xu et al., 2020). For details on the architecture training details, we refer to Section C.6.

For each architecture $a \in \mathcal{A}$, the dataset contains the following metrics: train/validation/test accuracy, training time and number of model parameters.

## C.3  DETAILS ON EACH OPTIMIZER

In this section we provide the hyperparameters used for the evaluations of NAS optimizers for the collection of our dataset. Many of the optimizers require a specialized representation to function on an architecture space because most of them are general HPO optimizers. As recently shown by White et al. (2020a), this representation can be critical for the performance of a NAS optimizer. Whenever the representation used by the Optimizer did not act directly on the graph representation, such as in RE, we detail how we represented the architecture for the optimizer. All optimizers were set to optimize the validation error.

**BANANAS.** We initialized BANANAS with 100 random architectures and modified the optimization of the surrogate model neural network, by adding early stopping based on a 90%/10% train/validation split and lowering the number of ensemble models to be trained from 5 to 3. These changes to bananas avoided a computational bottleneck in the training of the neural network.

**COMBO.** COMBO only attempts to maximize the acquisition function after the entire initial design (100 architectures) has completed. For workers which are done earlier, we sample a random architecture, hence increasing the initial design by the number of workers (30) we used for running the experiments. The search space considered in our work is larger than all search spaces evaluated in COMBO (Oh et al., 2019) and we regard not simply binary architectural choices, as we have to make choices about pairs of edges. Hence, we increased the number of initial samples for ascent acquisition function optimization from 20 to 30. Unfortunately, the optimization of the GP already became the bottleneck of the BO after around 600 function evaluations, leading to many workers waiting for new jobs to be assigned.

*Representation:* In contrast to the COMBO's original experimental setting, the DARTS search requires choices based on pairs of parents of intermediate nodes where the number of choices increase with the index of the intermediate nodes. The COMBO representation therefore consists of the graph cartesian product of the combinatorial choice graphs, increasing in size with each intermediate node. In addition, there exist 8 choices over the number of parameters for the operation in a cell.

**Differential Evolution.** DE was started with a generation size of 100. As we used a parallelized implementation, the workers would have to wait for one generation plus its mutations to be completed for selection to start. We decided to keep the workers busy by training randomly sampled architectures in this case, as random architectures provide us good coverage of the space. However, different methods using asynchronous DE selection would also be possible. Note, that the DE implementation by Awad et al. (2020), performs boundary checks and resamples components of any individual that exceeds 1.0. We use the rand1 mutation operation which generally favors exploration over exploitation.

*Representation:* DE uses a vector representation for each individual in the population. Categorical choices are scaled to lie within the unit interval [0, 1] and are rounded to the nearest category when converting back to the discrete representation in the implementation by Awad et al. (2020). Similarly to COMBO, we represent the increasing number of parent pair choices for the intermediate nodes by interpreting the respective entries to have an increasing number of sub-intervals in [0, 1].

**DARTS, GDAS, PC-DARTS and DrNAS.** We collected the architectures found by all of the above one-shot optimizers with their default search hyperparameters. We performed multiple searches for each one-shot optimizer.

**RE.** To allow for a good initial coverage before mutations start, we decided to randomly sample 3000 architectures as initial population. RE then proceeds with a sample size of 100 to extract well performing architectures from the population and mutates them. During mutations RE first decides whether to mutate the normal or reduction cell and then proceeds to perform either a parent change, an operation change or no mutation.

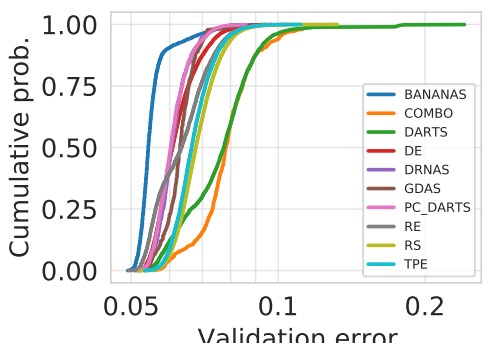

Figure 8: Empirical Cumulative Density Function (ECDF) plot comparing all optimizers in the dataset. Optimizers which cover good regions of the search space feature higher values in the low validation error region.

**TPE.** For TPE we use the default settings as also used by BOHB. We use the Kernel-Density-Estimator surrogate model and build two models where the good configs are chosen as the top 15%. The acquisition function's expected improvement is optimized by sampling 64 points.

## C.4 Optimizer Performance

The trajectories from the different NAS optimizers yield quite different performance distributions. This can be seen in Figure 8 which shows the ECDF of the validation errors of the architectures evaluated by each optimizer. As the computational budgets allocated to each optimizer vary widely, this data does not allow for a fair comparison between the optimizers. However, it is worth mentioning that the evaluations of BANANAS feature the best distribution of architecture performances, followed by PC-DARTS, DrNAS, DE, GDAS, and RE. TPE only evaluated marginally better architectures than RS, while COMBO and DARTS evaluated the worst architectures.

We also perform a t-SNE analysis on the data collected by the different optimizers in Figure 9. We find that RE discovers well-performing architectures which form clusters distinct from the architectures found via RS. We observe that COMBO searched previously unexplored areas of the search space. BANANAS, which found some of the best architectures, explores clusters outside the main cluster. However, it heavily exploits regions at the cost of exploration. We argue that this is a result of the optimization of the acquisition function via random mutations based on the previously found iterates, rather than on new random architectures. DE is the only optimizer which finds well performing architectures in the center of the embedding space.

## C.5 Cell topology, operations and noise

In this section, we investigate the influence of the cell topology and the operations on the performance of the architectures in our setting. The discovered properties of the search space inform our choice of metrics for the evaluation of different surrogate models.

First, we study how the validation error depends on the depth of architectures. Figure 10 visualizes the performance distribution of normal and reduction cells of different depth[5] by approximating empirical distributions with a kernel density estimation used in violin plots (Hwang et al., 1994). We observe that the performance distributions are similar for the normal and reduction cells with the same cell depth. Although cells of all depths can reach high performances, shallower cells seem slightly favored. Note that these observations are subject to changes in the hyperparameter setting, e.g. training for more epochs may render deeper cells more competitive. The best-found architecture features a normal and reduction cell of depth 4. Color-coding the cell depth in our t-SNE projection also confirms that the t-SNE analysis captures the cell depth well as a structural property (c.f. Figure 13). It also reinforces that the search space is well-covered.

We also show the distribution of normal and reduction cell depths of each optimizer in Figure 11 to get a sense for the diversity between the discovered architectures. We observe that DARTS and BANANAS generally find architectures with a shallow reduction cell and a deeper normal cell, while the reverse is true for RE. DE, TPE, COMBO and RS appear to find normal and reduction cells with similar cell depth.

Aside from the cell topology, we can also use our dataset to study the influence of operations to the architecture performance. The DARTS search space contains operation choices without parameters such as Skip-Connection, Max Pooling $3 \times 3$ and Avg Pooling $3 \times 3$. We visualize the influence of these parameter-free operations on the validation error in the normal and reduction cell in Figure 18a, respectively Figure 14. While pooling operations in the normal cell seem to have a negative impact on performance, a small number of skip connections improves the overall performance. This is somewhat expected, since the normal cell is dimension preserving and skip connections help training by improving

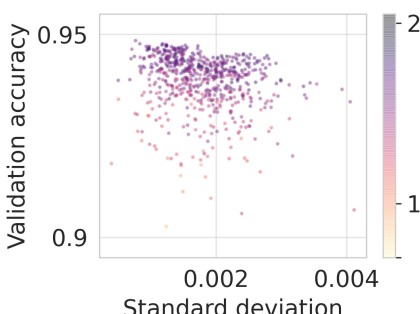

Figure 12: Standard deviation of the val. accuracy for multiple architecture evaluations.

gradient flow like in ResNets (He et al., 2016). In the reduction cell, the number of parameter-free operations has less effect as shown in Figure 14. In contrast to the normal cell where 2-3 skip-

---

[5]We follow the definition of cell depth used by Shu et al. (2020), i.e. the length of the longest simple path through the cell.

connections lead to generally better performance, the reduction cell shows no similar trend. For both cells, however, featuring many parameter-free operations significantly deteriorates performance. We therefore expect that a good surrogate also models this case as a poorly performing region.

### C.6    TRAINING DETAILS

Each architecture was evaluated on CIFAR-10 (Krizhevsky, 2009) using the standard 40k, 10k, 10k split for train, validation and test set. The networks were trained using SGD with momentum 0.9, initial learning rate of 0.025 and a cosine annealing schedule (Loshchilov & Hutter, 2017), annealing towards $10^{-8}$.

We apply a variety of common data augmentation techniques which differs from previous NAS benchmarks where the training accuracy of many evaluated architectures reached 100% (Ying et al., 2019; Dong & Yang, 2020) indicating overfitting on the training set. We used CutOut (DeVries & Taylor, 2017) with cutout length 16 and MixUp (Zhang et al., 2018) with alpha 0.2. For regularization, we used an auxiliary tower (Szegedy et al., 2015) with a weight of 0.4 and DropPath (Larsson et al., 2017) with drop probability of 0.2. We trained each architecture for 100 epochs with a batch

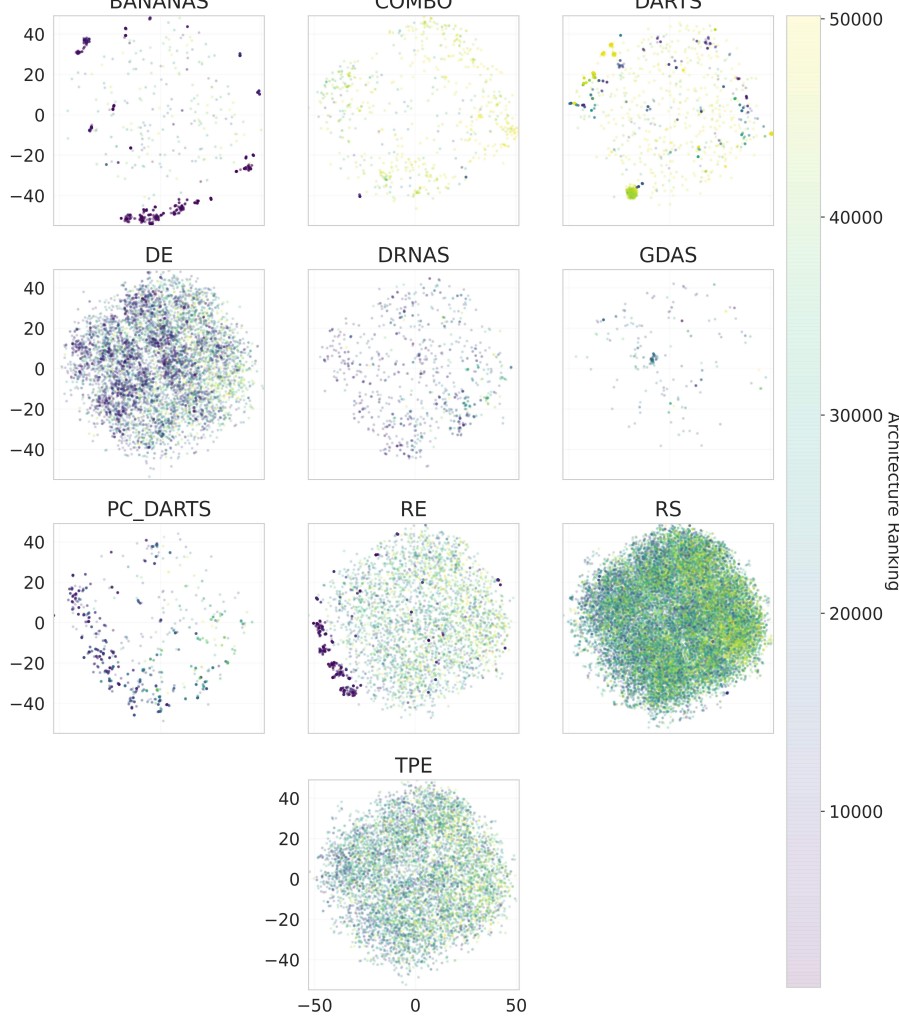

Figure 9: Visualization of the exploration of different parts of the architectural t-SNE embedding space for all optimizers used for data collection. The architecture ranking by validation accuracy (lower is better) is global over the entire data collection of all optimizers.

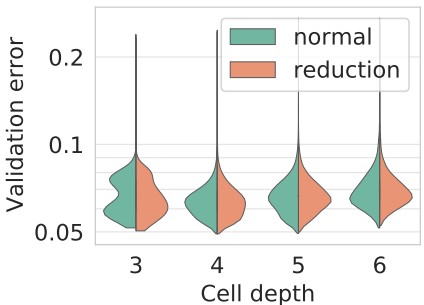
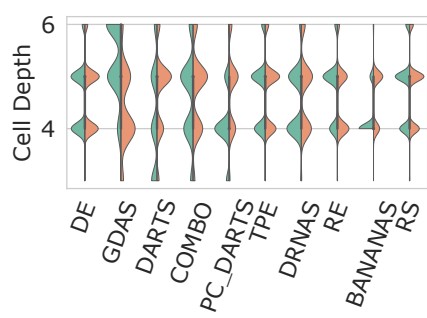

Figure 10: Distribution of the validation error for different cell depth.

Figure 11: Comparison between the normal and reduction cell depth for the architectures found by each optimizer.

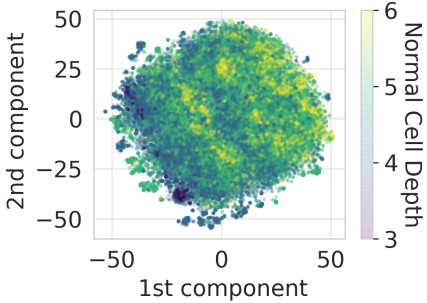
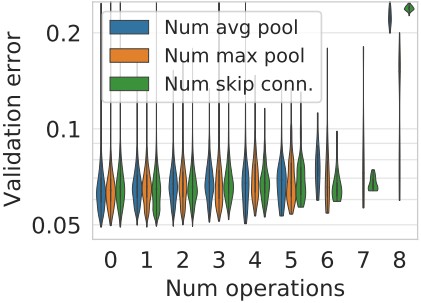

Figure 13: t-SNE projection colored by the depth of the normal cell.

Figure 14: Distribution of validation error in dependence of the number of parameter-free operations in the reduction cell. Violin plots are cut off at the respective observed minimum and maximum value.

size of 96, using 32 initial channels and 8 cell layers. We chose these values to be close to the proxy model used by DARTS while also achieving good performance.

## D  SURROGATE MODEL ANALYSIS

### D.1  PREPROCESSING OF THE GRAPH TOPOLOGY

**DGN preprocessing**   All DGN were implemented using PyTorch Geometric (Fey & Lenssen, 2019) which supports the aggregation of edge attributes. Hence, we can naturally represent the DARTS architecture cells, by assigning the embedded operations to the edges. The nodes are labeled as input, intermediate and output nodes. We represent the DARTS graph as shown in Figure 15, by connecting the output node of each cell type with the inputs of the other cell, allowing information from both cells to be aggregated per node during message passing. Note the self-loop on the output node of the normal cell, which we found necessary to get the best performance.

**Preprocessing for other surrogate models**   Since we make use of the framework implemented by BOHB (Falkner et al., 2018) to easily parallelize the architecture search algorithms across many compute nodes, we also represent our search space using ConfigSpace [6] (Lindauer et al., 2019). More precisely, we encode each pair of incoming edges for a cell as one choice of a categorical parameter. For instance, for node 4 in the normal cell, we add a parameter `inputs_node_normal_4` with the choices of edge pairs `0_1`, `0_2`, `0_3`, `1_2`, `1_3`, `2_3`. The edge operations are then implemented as categorical parameters for each edge and are only active if the corresponding edge was

---
[6] `https://github.com/automl/ConfigSpace`

chosen. For instance, in the example above, if the incoming edge 0 is sampled, the parameter associated with the edge from node 0 to node 4 becomes activate and one operation is sampled. We provide the configuration space with our code. For all non-DGN based surrogate models, we use the vector representation of a configuration given by ConfigSpace as input to the model. This vector representation contains one value between 0 and 1 for each parameter in the configuration space.

## D.2 DETAILS ON THE GIN

The GIN implementation on the Open Graph Benchmark (OGB) (Hu et al., 2020) uses virtual nodes (additional nodes which are connected to all nodes in the graph) to boost performance as well as generalization and consistently achieves good performance on their public leaderboards. Other GNNs from Errica et al. (2020), such as DGCNN and DiffPool, performed worse in our initial experiments and are therefore not considered.

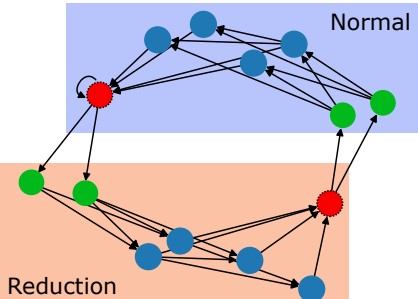

Figure 15: Architecture with inputs in green, intermediate nodes in blue and outputs in red.

Following recent work in Predictor-based NAS (Ning et al., 2020; Xu et al., 2019b), we use a per batch ranking loss because the ranking of an architecture is equally important to an accurate prediction of the validation accuracy in a NAS setting. We use the ranking loss formulation by GATES (Ning et al., 2020) which is a hinge pair-wise ranking loss with margin m=0.1.

## D.3 DETAILS ON HPO

All detailed table for the hyperparameter ranges for the HPO and the best values found by BOHB are listed in Table 7.

## D.4 HPO FOR RUNTIME PREDICTION MODEL

Our runtime prediction model is an LGB model trained on the runtimes of architecture evaluations of DE. This is because we partially evaluated the architectures utilizing different CPUs. Hence, we only choose to train on the evaluations carried out by the same optimizer on the same hardware to keep a consistent estimate of the runtime. DE is a good choice in this case because it both explored and exploited the architecture space well. The HPO space used for the LGB runtime model is the same used for the LGB surrogate model.

## D.5 LEAVE ONE-OPTIMIZER-OUT ANALYSIS

A detailed scatter plot of the predicted performance against the true performance for each optimizer and surrogate model in an LOOO analysis is provided in Figure 16 and Figure 17.

## D.6 PARAMETER-FREE OPERATIONS

Several works have found that methods based on DARTS (Liu et al., 2019b) are prone to finding sub-optimal architectures that contain many, or even only, parameter-free operations (max. pooling, avg. pooling or skip connections) and perform poorly (Zela et al., 2020a). We therefore evaluated the surrogate models on such architectures by replacing a random selection of operations in a cell with one type of parameter-free operations to match a certain ratio of parameter-free operations in a cell. This analysis is carried out over the test set of the surrogate models and hence contains architectures collected by all optimizers. For a more robust analysis, we repeated this experiment 4 times for each ratio of operations to replace.

**Results** Figure 18 shows that both the GIN and the XGB model correctly predict that the accuracy drops with too many parameter-free operations, particularly for skip connections. The groundtruth of architectures with only parameter-free operations is displayed as scatter plot. Out of the two models,

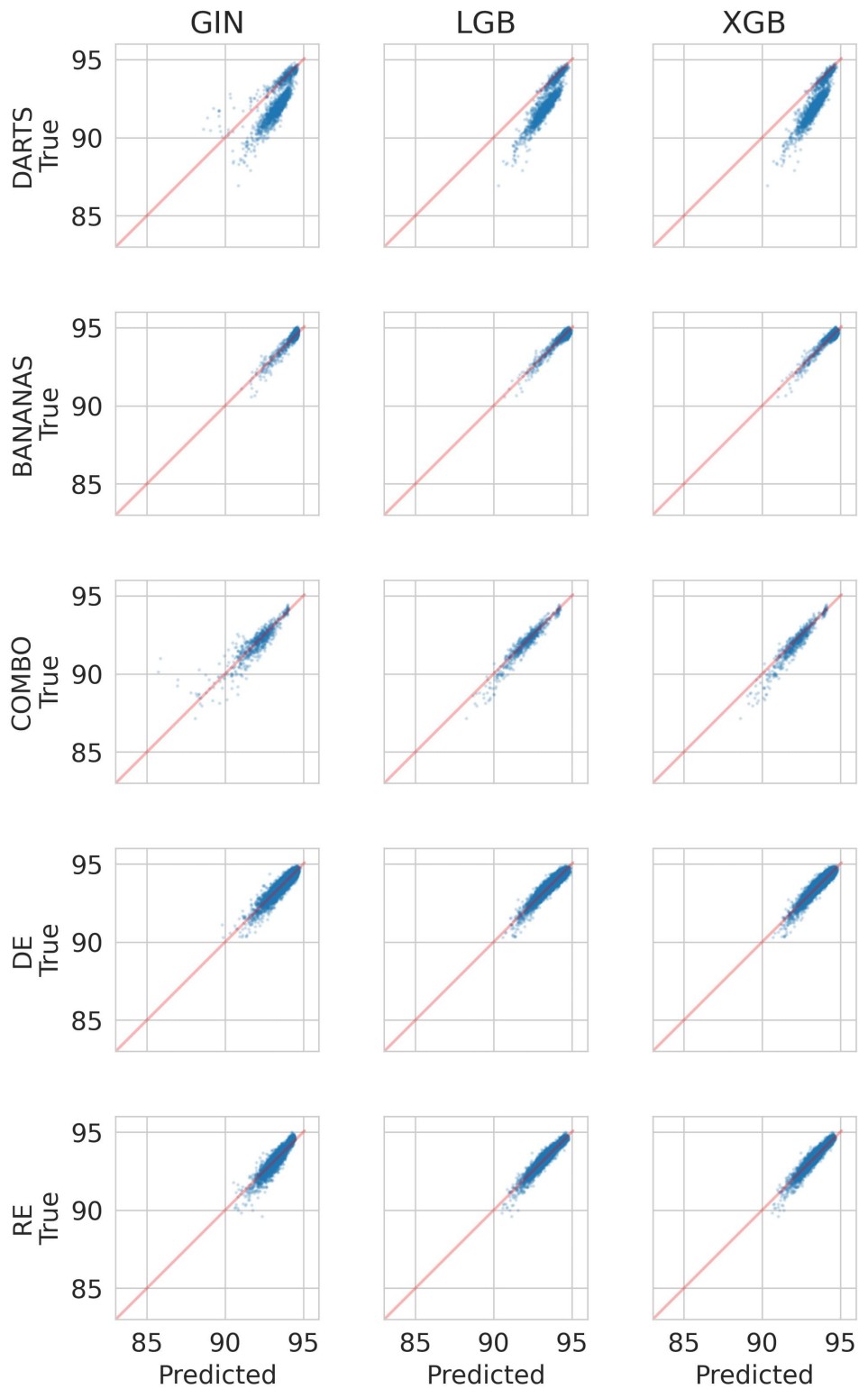

Figure 16: Scatter plots of the predicted performance against the true performance of different surrogate models on the test set in a Leave-One-Optimizer-Out setting.

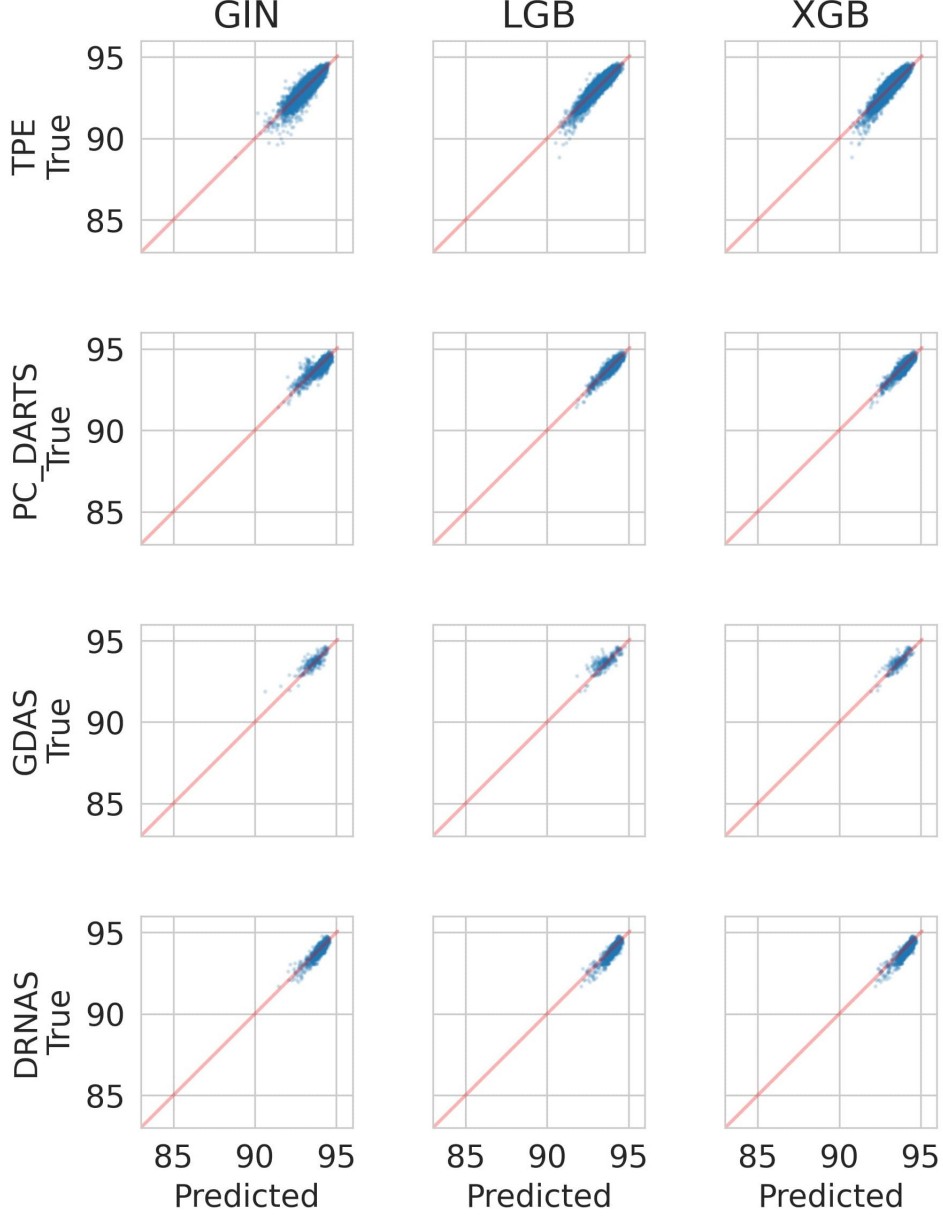

Figure 17: (continued) Scatter plots of the predicted performance against the true performance of different surrogate models on the test set in a Leave-One-Optimizer-Out setting.

| Model | Hyperparameter | Range | Log-transform | Default Value |
|---|---|---|---|---|
| GIN | Hidden dim. | [16, 256] | true | 24 |
| | Num. Layers | [2, 10] | false | 8 |
| | Dropout Prob. | [0, 1] | false | 0.035 |
| | Learning rate | [1e-3, 1e-2] | true | 0.0777 |
| | Learning rate min. | const. | - | 0.0 |
| | Batch size | const. | - | 51 |
| | Undirected graph | [true, false] | - | false |
| | Pairwise ranking loss | [true, false] | - | true |
| | Self-Loops | [true, false] | - | false |
| | Loss log transform | [true, false] | - | true |
| | Node degree one-hot | const. | - | true |
| BANANAS | Num. Layers | [1, 10] | true | 17 |
| | Layer width | [16, 256] | true | 31 |
| | Dropout Prob. | const. | - | 0.0 |
| | Learning rate | [1e-3, 1e-1] | true | 0.0021 |
| | Learning rate min. | const. | - | 0.0 |
| | Batch size | [16, 128] | - | 122 |
| | Loss log transform | [true, false] | - | true |
| | Pairwise ranking loss | [true, false] | - | false |
| XGBoost | Early Stopping Rounds | const. | - | 100 |
| | Booster | const. | - | gbtree |
| | Max. depth | [1, 15] | false | 13 |
| | Min. child weight | [1, 100] | true | 39 |
| | Col. sample bylevel | [0.0, 1.0] | false | 0.6909 |
| | Col. sample bytree | [0.0, 1.0] | false | 0.2545 |
| | lambda | [0.001, 1000] | true | 31.3933 |
| | alpha | [0.001, 1000] | true | 0.2417 |
| | Learning rate | [0.001, 0.1] | true | 0.00824 |
| LGBoost | Early stop. rounds | const. | - | 100 |
| | Max. depth | [1, 25] | false | 18 |
| | Num. leaves | [10, 100] | false | 40 |
| | Max. bin | [100, 400] | false | 336 |
| | Feature Fraction | [0.1, 1.0] | false | 0.1532 |
| | Min. child weight | [0.001, 10] | true | 0.5822 |
| | Lambda L1 | [0.001, 1000] | true | 0.0115 |
| | Lambda L2 | [0.001, 1000] | true | 134.5075 |
| | Boosting type | const. | - | gbdt |
| | Learning rate | [0.001, 0.1] | true | 0.0218 |
| Random Forest | Num. estimators | [16, 128] | true | 116 |
| | Min. samples split. | [2, 20] | false | 2 |
| | Min. samples leaf | [1, 20] | false | 2 |
| | Max. features | [0.1, 1.0] | false | 0.1706 |
| | Bootstrap | [true, false] | - | false |
| $\epsilon$-SVR | C | [1.0, 20.0] | true | 3.066 |
| | coef. 0 | [-0.5, 0.5] | false | 0.1627 |
| | degree | [1, 128] | true | 1 |
| | epsilon | [0.01, 0.99] | true | 0.0251 |
| | gamma | [scale, auto] | - | auto |
| | kernel | [linear, rbf, poly, sigmoid] | - | sigmoid |
| | shrinking | [true, false] | - | false |
| | tol | [0.0001, 0.01] | - | 0.0021 |
| $\mu$-SVR | C | [1.0, 20.0] | true | 5.3131 |
| | coef. 0 | [-0.5, 0.5] | false | -0.3316 |
| | degree | [1, 128] | true | 128 |
| | gamma | [scale, auto] | - | scale |
| | kernel | [linear, rbf, poly, sigmoid] | - | rbf |
| | nu | [0.01, 1.0] | false | 0.1839 |
| | shrinking | [true, false] | - | true |
| | tol | [0.0001, 0.01] | - | 0.003 |

Table 7: Hyperparameters of the surrogate models and the default values found via HPO.

XGB captures the slight performance improvement of using a few skip connections better. LGB failed to capture this trend but performed very similarly to XGB for the high number of parameter-free operations.

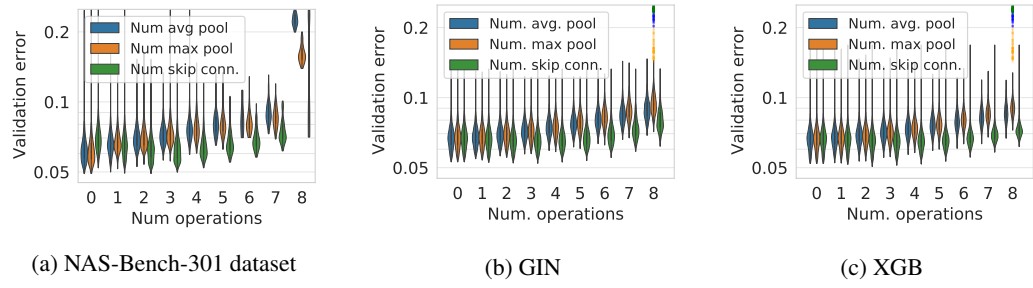

Figure 18: (Left) Distribution of validation error in dependence of the number of parameter-free operations in the normal cell on the NAS-Bench-301 dataset. (Middle and Right) Predictions of the GIN and XGB surrogate model. The collected groundtruth data is shown as scatter plot. Violin plots are cut off at the respective observed minimum and maximum value.

### D.7 CELL TOPOLOGY ANALYSIS

Furthermore, we analyze how accurate changes in the cell topology (rather than in the operations) are modeled by the surrogates. We collected groundtruth data by evaluating all $\prod_{k=1}^{4} \frac{(k+1)k}{2} = 180$ different cell topologies (not accounting for isomorphisms) with fixed sets of operations. We assigned the same architecture to the normal and reduction cell, to focus on the effect of the cell topology. We sampled 10 operation sets uniformly at random, leading to 1800 architectures as groundtruth for this analysis.

We evaluated all architectures and group the results based on the cell depth. For each of the possible cell depths, we then computed the sparse Kendall $\tau$ rank correlation between the predicted and true validation accuracy.

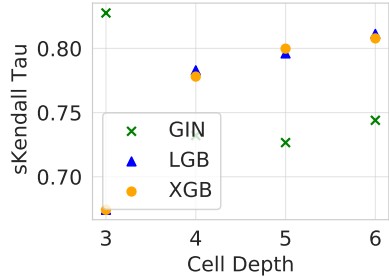

Figure 19: Comparison between GIN, XGB and LGB in the cell topology analysis.

**Results** Results of the cell topology analysis are shown in Figure 19. We observe that LGB slightly outperforms XGB, both of which perform better on deeper cells. The GIN performs best for the shallowest cells.

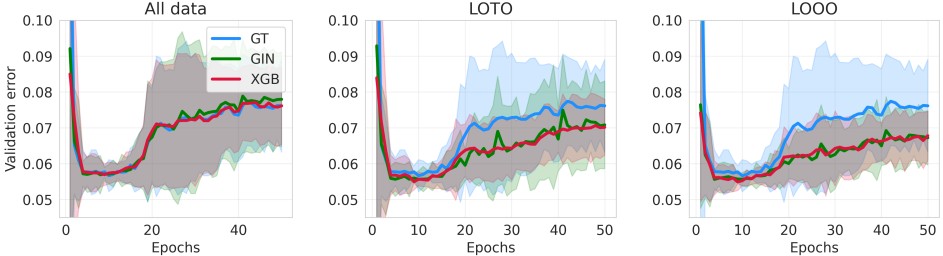

Figure 20: Ground truth (GT) and surrogate trajectories on a constrained search space where the surrogates are trained with all data, leaving out the trajectories under consideration (LOTO), and leaving out all DARTS architectures (LOOO).

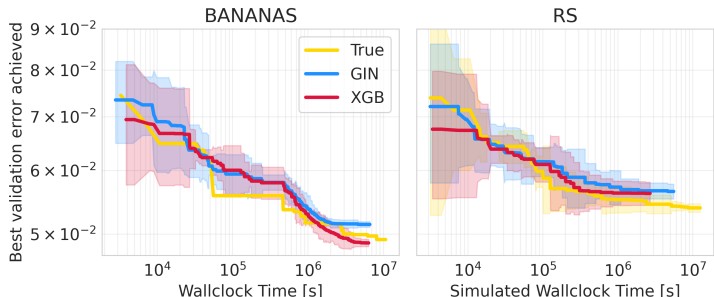

Figure 22: Comparison between the observed true trajectory of BANANAS and RS with the surrogate benchmarks only trained on well performing regions of the space

## E    BENCHMARK ANALYSIS

### E.1    ONE-SHOT TRAJECTORIES

To obtain groundtruth trajectories for DARTS, PC-DARTS and GDAS, we performed 5 runs for each optimizer with 50 search epochs and evaluated the architecture obtained by discretizing the one-shot model at each search epoch. For DARTS, in addition to the default search space, we collected trajectories on the constrained search spaces from Zela et al. (2020a) to cover a failure case where DARTS diverges and finds architectures that only contain skip connections in the normal cell. To show that our benchmark is able to predict this divergent behavior, we show surrogate trajectories when training on all data, when leaving out the trajectories under consideration from the training data, and when leaving out all DARTS data in Figure 20.

While the surrogates model the divergence in all cases, they still overpredict the architectures with only skip connections in the normal cell especially when leaving out all data from DARTS. The bad performance of these architectures is predicted more accurately when including data from other DARTS runs. This can be attributed to the fact that the surrogate models have not seen any, respectively very few data, in this region of the search space. Nevertheless, it is modeled as a bad-performing region and we expect that this could be further improved on by including additional training data accordingly, since including all data in training shows that the models are capable to of capturing this behavior.

### E.2    ABLATION STUDY: FITTING SURROGATE MODELS ONLY ON WELL-PERFORMING REGIONS OF THE SEARCH SPACE

To assess whether poorly-performing architectures are important for the surrogate benchmark, we fitted a GIN ensemble and an XGB ensemble model only on architectures that achieved a validation accuracy above 92%. We then tested on all architectures that achieved a validation below 92%.

Indeed, we observe that the resulting surrogate model overpredicts accuracy in regions of the space with poor performance, resulting in a low $R^2$ of -0.142 and sparse Kendall tau of 0.293 for the GIN. The results for one member of the GIN ensemble are shown in Figure 21. The XGB model achieved similar results. Next, to study whether these weaker surrogate models can still be used to benchmark NAS optimizers, we also studied optimization trajectories of NAS optimizers on surrogate bench-

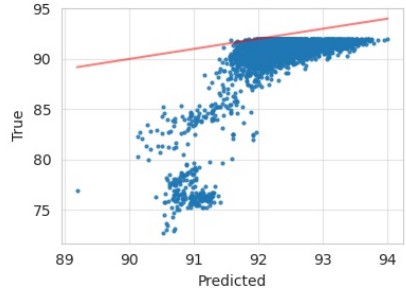

Figure 21: Scatter plot of GIN predictions on architectures that achieved below 92% validation accuracy.

marks based on these surrogate models. Figure 22 shows that these surrogate models indeed suffice to accurately predict the performance achieved by Random Search and BANANAS as a function of time.

### E.3 Ablation Study: Fitting surrogate models only with random data

In this section, we would like to take the Leave-One-Optimizer-Out analysis from Section 5.1 one step further by leaving out *all* architectures that were collected from NAS optimizers other than random search. While the LOOO analysis removes some "bias" from the benchmark ("bias" referring to its precision in a subspace), there still is the possibility that different optimizers we used explore similar subspaces, and leaving out one of them still yields "bias" induced by architectures from a similar optimizer used for generating training data. For instance, the t-SNE analysis from Figure 9 suggests that some optimizers exploit very distinct regions (e.g., BANANAS and DE) while others exploit regions somewhat similar to others (e.g., RE and PC-DARTS). The exploration behavior, on the other hand, is quite similar across optimizers since most of them perform random sampling in the beginning. Thus, in the following, we investigate whether we can create a benchmark that has no prior information about solutions any optimizer might find.

To that end, we studied surrogate models based i) only on the 23746 architectures explored by random search and ii) only on 23 746 (47.3%) architectures of the original training set (sampled in a stratified manner, i.e., using 47.3% of the architectures from each of our sources of architectures).

First, we investigated the difference in the predictive performance of surrogates based on these two different types of architectures. Specifically, we fitted our GNN and XGB surrogate models on different subsets of the respective training sets and assess their predictions on unseen architectures from all optimziers as a test set. Figure 23 shows that including architectures from optimizer trajectories in the training set consistently yields significantly better generalization.

Next, we also studied the usefulness of surrogate benchmarks based on the 23 746 random architectures, compared to surrogate benchmarks based on the 23 746 architectures sampled in a stratified manner from the original set of architectures. Specifically, we used them to assess the best performance achieved by various NAS optimizers as a function of time. Comparing the trajectories in Figure 24 (based on purely random architectures for training) and Figure 25 (based on 23 746 architectures sampled in a stratified manner), we find that the surrogates fitted only on random architectures work just as well for this task as the surrogates that use architectures from NAS optimizers in their training set.

Given this positive result for surrogates based purely on random architectures, we conclude that it is indeed possible to create surrogate NAS benchmarks that are by design free of bias towards any particular NAS optimizer (other than random search). While the inclusion of architectures generated with NAS optimizers in the training set substantially improves performance predictions of individual architectures, realistic trajectories of incumbent performance as a function of time can also be obtained with surrogate benchmarks based solely on random architectures. We note that the "unbiased" benchmark could possibly be further improved by utilizing more sophisticated space-filling sampling methods, such as the ones mentioned in Appendix C.2, or by deploying surrogate models that extrapolate well.

## F Guidelines for Creating Surrogate Benchmarks

In order to help with the design of realistic surrogate benchmarks in the future, we provide the following list of guidelines:

- Data Collection: The data collected for the NAS benchmark should provide (1) a good overall coverage, (2) explore strong regions of the space well, and (3) optimally also cover special areas in which poor generalization performance may otherwise be expected. We would like to stress that depending on the search space, a good overall coverage may already be sufficient to correctly assess the ranking of different optimizers, but as shown in Appendix E.3 additional architectures from strong regions of the space allow to increase the fidelity of the surrogate model.
  1. A good overall coverage can be obtained by random search (as in our case), but one could also imagine using better space-filling designs or adaptive methods for covering the space even better. In order to add additional varied architectures, one could also think about fitting one or more surrogate models to the data collected thus far, finding the regions of maximal predicted uncertainty, evaluate architectures there and add them to the collected data, and iterate. This would constitute an active learning approach.

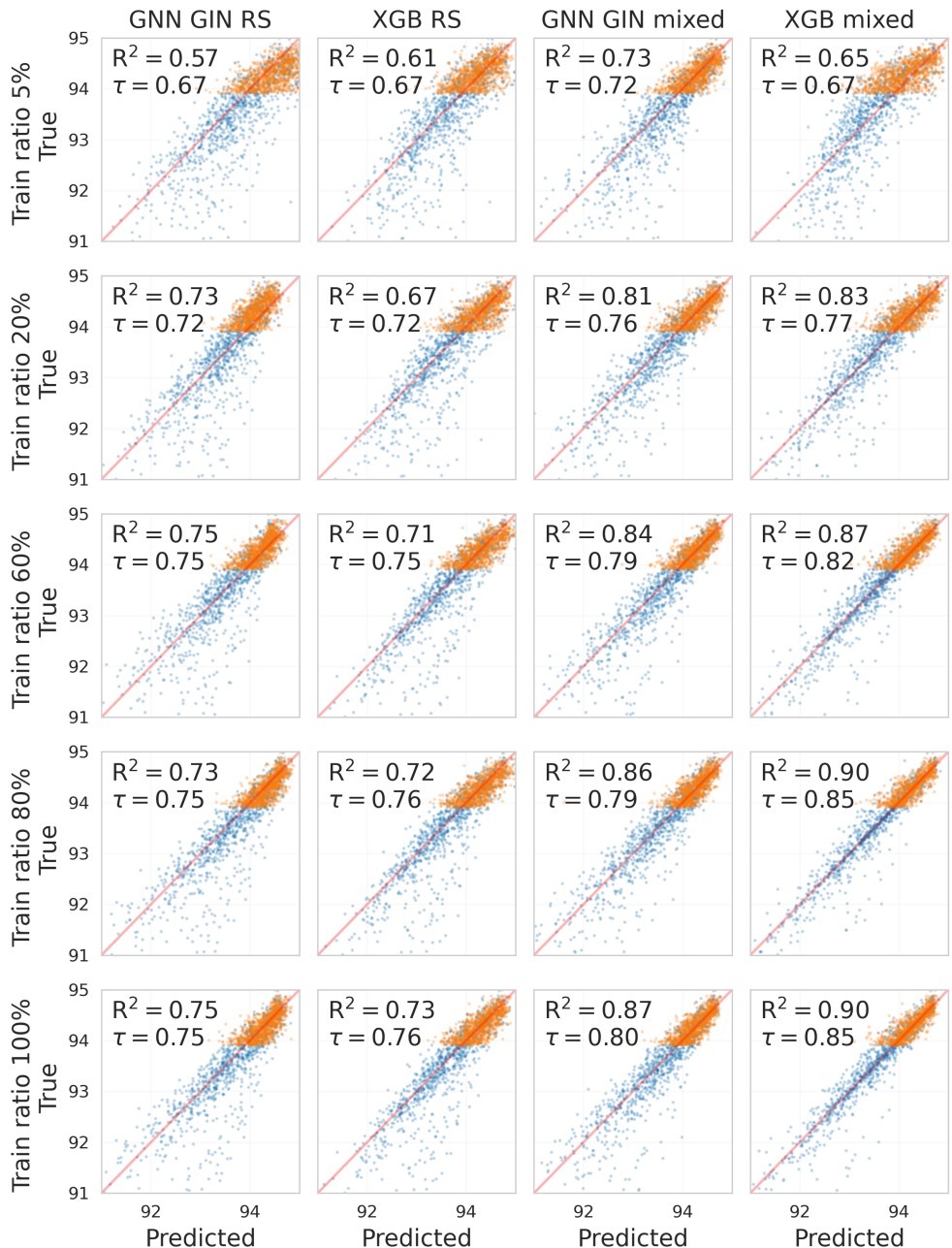

Figure 23: Scatter plots of the predicted performance against the true performance of the GNN GIN/XGB surrogate models trained with different ratios of training data. "RS" indicates that the training set only includes architectures from random search, "mixed" indicates the training set includes architectures from all optimizers. Training set sizes are identical for the two cases. The test set contains architectures from all optimizers. For better display, we show 1000 randomly sampled architectures (blue) and 1000 architectures sampled from the top 1000 architectures (orange). For each case we also show the $R^2$ and Kendall-$\tau$ coefficients on the whole test set.

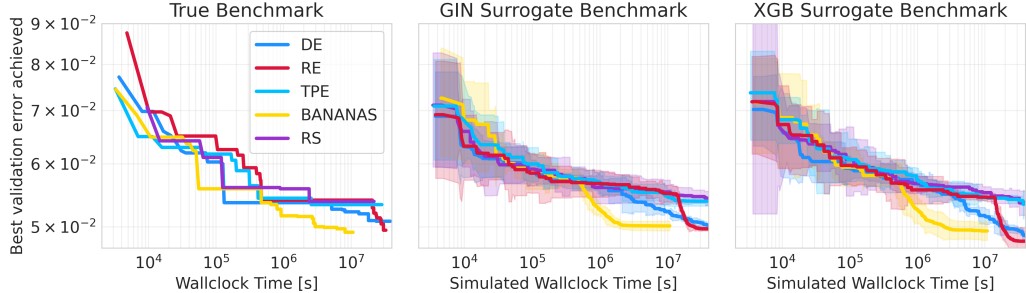

Figure 24: Anytime performance of different optimizers on the real benchmark (left) and the surrogate benchmark (GIN (middle) and XGB (right)) when training ensembles only on data collected by random search. Trajectories on the surrogate benchmark are averaged over 5 optimizer runs.

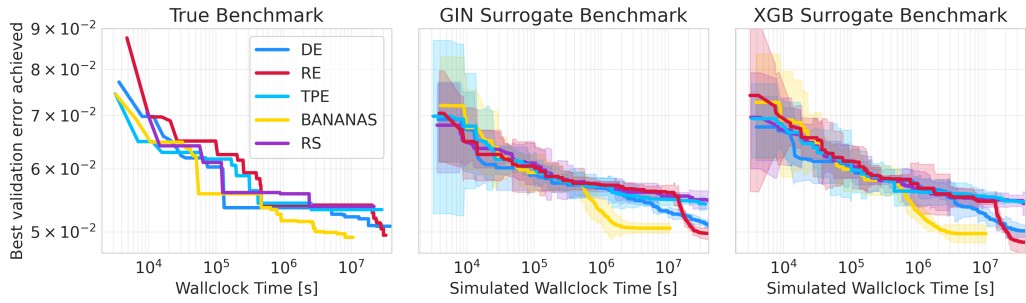

Figure 25: Anytime performance of different optimizers on the real benchmark (left) and the surrogate benchmark (GIN (middle) and XGB (right)) when training ensembles on 47.3% of the data collected from all optimizers. Trajectories on the surrogate benchmark are averaged over 5 optimizer runs.

2. A convenient and efficient way to identify regions of strong architectures is to run NAS methods. In this case, the found regions should not only be based on the strong architectures one NAS method finds but rather on a set of strong and varied NAS methods (such as, in our case, one-shot methods and different types of discrete methods, such as Bayesian optimization and evolution). In order to add additional strong architectures, one could also think about fitting one or more several surrogate models to the data collected thus far, finding the predicted optima of these models, evaluate and add them to the collected data and iterate. This would constitute a special type of Bayesian optimization.

3. Special areas in which poor generalization performance may otherwise be expected may, as in our case, e.g., include architectures with many parameterless connections, and in particular, skip connections. Other types of failure modes the community learns about would also be useful to cover.

- Surrogate Models: As mentioned in the guidelines for using a surrogate benchmark (see Section 7), benchmarking an algorithm that internally uses the same model type as the surrogate model should be avoided. Therefore, to provide a benchmark for a diverse set of algorithms, we recommend providing different types of surrogate models with a surrogate benchmark. Also, in order to guard against a possible case of "bias" in a surrogate benchmark (in the sense of making more accurate predictions for architectures explored by a particular type of NAS optimizer), we recommend to provide two versions of a surrogate: one based on all available training architectures (including those found by NAS optimizers), and one based only on the data gathered for overall coverage (1. above).

- Verification: As a means to verify surrogate models, we stress the importance of leave-one-optimizer-out experiments both for data fit and benchmarking, which simulate the benchmarking of 'unseen' optimizers.

- Since most surrogate benchmarks will continue to grow for some time after their first release, to allow apples-to-apples comparisons, we strongly encourage to only release surrogate benchmarks with a version number.

- In order to allow the evaluation of multi-objective NAS methods, we encourage the logging of as many relevant metrics of the evaluated architectures other than accuracy as possible, including training time, number of parameters, and multiply-adds.

- Alongside a released surrogate benchmark, we strongly encourage to release the training data its surrogate(s) were constructed on, as well as the test data used to validate it.

- In order to facilitate checking hypotheses gained using the surrogate benchmarks in real experiments, the complete source code for training the architectures should be open-sourced alongside the repository, allowing to easily go back and forth between querying the model and gathering new data.

