# OpenReview forum: "NAS-Bench-301 and the Case for Surrogate Benchmarks for Neural Architecture Search"
_ICLR.cc/2021/Conference — Reject_

### Official Review · AnonReviewer2 · 2020-10-16

**Rating:** 3
**Confidence:** 5

**Review:**

In this work, the authors train a model on a subset of architectures (~60k) in the DARTS search space and use this model to predict the performance of architectures outside of that subset in DARTS. They then use this as a surrogate for having to perform network training for evaluating NAS algorithms.

NAS benchmarks are a good thing, given that they facilitate NAS research outside of labs with lots of resources. Saying this, I think there are serious issues with this paper.

NAS-Bench-301 is a model that predicts the performance of networks in the DARTS search space. We know from https://arxiv.org/abs/1912.12522v3 that there is a significant lack of variety in the performance of models within this space, and this work compounds that. Additionally, this paper only considers CIFAR-10. This is a step back from NAS-Bench-201 which despite its small size, did contain multiple datasets. The compute used by the authors (training 60k networks) has gone into differentiating a bunch of networks that are all quite good, within a few percentage points of error. I’m not sure how this is of practical use.

A problem with NAS-Bench-101 and 201 is that the search spaces are small (423k, 6k) as the authors point out. NAS-Bench-301 encapsulates the DARTS search space which is much bigger (10^18). However, from https://arxiv.org/abs/1912.12522v3 we see that randomly sampling within this space gives networks between 96.5% and 97.5%. I would argue that it doesn’t matter how large a space is if it is lacking in variety; Every possible network works well enough. In 101/201 we see networks across a much larger range (typically between 80-95% although there are some much lower). I believe this is more interesting from a research perspective as we would like to apply NAS to situations where networks can break (and avoid this happening). The DARTS networks do explore a slightly higher accuracy range but it is not at state-of-art levels, so the added value is not clear.

The authors are critical of random search, stating (i) `random search stagnates and cannot identify one of the best architectures even after tens of thousands of evaluations`, (ii) NAS-Bench-201 (Dong & Yang, 2020) only contains 6466 unique architectures in total, causing the median of random search runs to find the best architecture after only 3233 evaluations.  However, (i) makes random search sound like it is failing, where on Figure 4 we can see at 10^4 (s) random search is doing very well  - matching or beating all the other techniques.  We know from https://arxiv.org/abs/1806.09055  that random search works well on the DARTS  space (probably due to the lack of variety). (ii) Random search does work well on NAS-Bench-201 but the DARTS algorithm fails, even though it works on this space. The statement in the abstract that using previous benchmarks  “leads to unrealistic results that do not transfer to larger search spaces.” does not tell the whole story, as there appears to be more going on than just the size of the search space. Some results on this large DARTS search space, do not transfer to the smaller search spaces. It seems that there is more to a search space than the number of architectures within it.

In terms of presentation, this paper is well written, although the figures could be larger. I appreciate that this is problem to keep within the page limit.

NAS benchmark papers are important and shape the direction of research in the field. This paper doesn’t present a benchmark. It provides a model that represents computationally-efficient means of getting network accuracies from the DARTS search space. I appreciate this endeavour and it could be of use outside of this space — being able to map a 10^18 space with 50000 points indicates that the effective size of the space is much smaller, and is highly predictable.

This space, although large, has very little variety in terms of network accuracy. I believe NAS-Bench-101 and 201 despite their sizes represent more varied search spaces. 201 also covers multiple datasets, whereas 301 only has CIFAR-10 (which it feels like we are saturating on).  I recommend rejection for this paper, as I do not believe it represents a step forward in the way we benchmark our NAS algorithms. We need to develop more interesting search spaces,  rather than advocating exploration of uninteresting ones.

---

> ### Author Response · Authors · 2020-11-16
> **Response to AnonReviewer2 (Part 1/3)**
>
> We would like to thank the reviewer for taking the time to read our paper and for his/her feedback. We now address the reviewer’s concerns:
>
> 1. **We know from https://arxiv.org/abs/1912.12522v3 that there is a significant lack of variety in the performance of models within this space, and this work compounds that.**: We note that the distribution of architecture performances in the DARTS search space also depends on the choice of macro architecture and training pipeline [1]. We would also like to point out that Figure 8 in the appendix shows the ECDF for different optimizers and indicates that there is variety in our setting. We agree with the reviewer that the DARTS search space does not yield the highest variety and that the NAS research community *must* move forward to other interesting spaces. Nevertheless, the DARTS space is still the most widely used non-tabular search space and many NAS papers are burning substantial GPU time on this space. With NAS-Bench-301, we provide a cheap way to prototype and test NAS algorithms that we hope will be of great help to the community, particularly practitioners with fewer resources. While tabular benchmarks restrict the community to small search spaces that are only feasible to evaluate exhaustively, we believe that NAS-Bench-301 might be a milestone for creating future fast-to-evaluate surrogate benchmarks on large and interesting search spaces, enabling the community to focus its resources and time on these new benchmarks rather than spending them on the DARTS space.
>
> 2. **Additionally, this paper only considers CIFAR-10. This is a step back from NAS-Bench-201 which despite its small size, did contain multiple datasets.**: As mentioned above, we chose our search space & dataset, DARTS + CIFAR-10, based on the most popular space for which most computational resources are currently used. Unfortunately, our own computational budget did not allow us to cover multiple datasets but our approach trivially applies to other spaces and datasets. We would also like to emphasize that our work is the first to propose surrogate benchmarks for NAS and shows that they are indeed applicable. In doing so, our fundamental work on the methodology for creating fast-to-evaluate NAS benchmarks opens up the door to fast-to-evaluate NAS benchmarks for the large set of varied and exciting NAS applications the community is developing these days.
>
> 3. **The compute used by the authors (training 60k networks) has gone into differentiating a bunch of networks that are all quite good, within a few percentage points of error. I’m not sure how this is of practical use.**: We point to Figure 8 in our Appendix which shows ECDFs for different optimizers. We note that there are architectures with performances between 80-95% validation accuracy. Importantly, NAS-Bench-101 and NAS-Bench-201 report the performance of the architectures in the whole search space, while we focus more on collecting architectures from distinct regions of the search space via several NAS optimizers. We refer the reviewer to the t-SNE plots in Fig. 9 (Appendix C.4) to visualize that these optimizers exploit different well-performing regions of the architecture space (e.g. compare DE and BANANAS).

---

> > ### Author Response · Authors · 2020-11-16
> > **Response to AnonReviewer2 (Part 2/3)**
> >
> > 4. **NAS-Bench-301 encapsulates the DARTS search space which is much bigger (10^18). However, from https://arxiv.org/abs/1912.12522v3 we see that randomly sampling within this space gives networks between 96.5% and 97.5%. I would argue that it doesn’t matter how large a space is if it is lacking in variety; Every possible network works well enough. In 101/201 we see networks across a much larger range (typically between 80-95% although there are some much lower). I believe this is more interesting from a research perspective as we would like to apply NAS to situations where networks can break (and avoid this happening). The DARTS networks do explore a slightly higher accuracy range but it is not at state-of-art levels, so the added value is not clear.**: We agree that random sampling yields decent performances with a *heavily optimized training pipeline* as in [1]. However, this is not the case with the original DARTS training pipeline, which we use. Randomly sampling an architecture in the DARTS space using our training pipeline yields a test error of more than 7\% as shown in Fig. 4 (left plot; beginning of the curves where every optimizer randomly samples initially), while the best found architectures have a test error below 5\%. In [1], going more and more towards state-of-the-art performance by adding various regularization techniques or using optimized hyperparameter settings on a large macro architecture and eventually this yields diminishing returns for most of the architectures and most of them fall into the same good range of performance. This is clearly not the case in NAS-Bench-301 (and also for NAS-Bench-101 and 201), where we can clearly distinguish a random sample and an optimized architecture (more than 2\% error difference). It would also be interesting in our opinion to increase the macro architecture size and use a better training pipeline to train the architectures in NAS-Bench-101 or 201 to see if there will be as much variability as they have with the used network sizes and training pipelines.
> >
> > 5. **However, (i) makes random search sound like it is failing, where on Figure 4 we can see at 10^4 (s) random search is doing very well - matching or beating all the other techniques. We know from https://arxiv.org/abs/1806.09055 that random search works well on the DARTS space (probably due to the lack of variety).**:
> > By design, random search will not fail, but indeed, as also observed in many HPO benchmarks it performs well especially in the beginning of optimization where optimizers typically need explorative behaviour (and are thus very similar to random search), while the advantages of more guided optimizers are more apparent in later stages of optimization. There, more guided NAS optimizers succeed in exploiting good areas in the space, whereas random search struggles to find the best architectures. For instance, mentally drawing a horizontal line through Figure 4 at 5% error shows that BANANAS is orders of magnitude faster in finding these best architectures.
> >
> > 6. **(ii) Random search does work well on NAS-Bench-201 but the DARTS algorithm fails, even though it works on this space. The statement in the abstract that using previous benchmarks “leads to unrealistic results that do not transfer to larger search spaces.” does not tell the whole story, as there appears to be more going on than just the size of the search space. Some results on this large DARTS search space, do not transfer to the smaller search spaces. It seems that there is more to a search space than the number of architectures within it.**: We agree that that statement might have been confusing; in making it, we had our results for Local Search in mind (the results we show in Section 6), not random search. For Local Search [2], performances on the small spaces of NAS-Bench-101 and NAS-Bench-201 are qualitatively different than for the large space of NAS-Bench-301. As we show in our Section 6, Local Search actually does catch with other approaches when given enough time, but its initial performance is much worse than on the small benchmarks.
> > Another point on which we agree with the reviewer is the fact that many one-shot NAS optimizers, such as DARTS, do not work well on smaller search spaces, even though they do on the original DARTS space. This is carefully investigated in [3], where the authors show that even on sub-spaces of the original DARTS space, the DARTS algorithm can fail. Note that in NAS-Bench-301 these sub-spaces are included and the surrogate models can model these failure modes effectively (as shown in Appendix E.1 and Figure 19).

---

> > > ### Author Response · Authors · 2020-11-16
> > > **Response to AnonReviewer2 (Part 3/3)**
> > >
> > > 7. **In terms of presentation, this paper is well written, although the figures could be larger. I appreciate that this is problem to keep within the page limit.**: Thank you, we have updated figure sizes within the new page limit.
> > >
> > > 8. **NAS benchmark papers are important and shape the direction of research in the field. This paper doesn’t present a benchmark.**: As we show in our Section 2, a surrogate model can yield a benchmark which is as accurate or even more accurate than a tabular benchmark. NAS-Bench-301 is a benchmark which is based on surrogate predictions rather than tabular entries. Our concept of a NAS benchmark consists of a search space, a dataset and a fixed training pipeline used to train the architectures in the space and NAS-Bench-301 features all these attributes.
> > >
> > > 9. **This space, although large, has very little variety in terms of network accuracy. I believe NAS-Bench-101 and 201 despite their sizes represent more varied search spaces. 201 also covers multiple datasets, whereas 301 only has CIFAR-10 (which it feels like we are saturating on). I recommend rejection for this paper, as I do not believe it represents a step forward in the way we benchmark our NAS algorithms. We need to develop more interesting search spaces, rather than advocating exploration of uninteresting ones.**: We strongly agree with the reviewer that we need more interesting search spaces in order to advance the field. However, we strongly disagree that NAS-Bench-301 is a step backwards in the way we benchmark NAS algorithms. Here are the reasons: To the best of our knowledge, every new published NAS method for image classification in the last 2-3 years evaluates on this search space and CIFAR-10. We do not want to argue that this is a good practice and it is true that we might be saturating on this particular space. However, NAS practitioners still want (have) to evaluate on this space and dataset and this requires a lot of resources and time (not mentioning the time required for prototyping). We hope that via NAS-Bench-301 we 1) will ameliorate this heavy cost of prototyping and evaluating new NAS methods on this space (this is also desirable for environmental reasons); and 2) provide a practical methodology and succinct guidelines on how to construct surrogate benchmarks. Given enough compute power, one can do exactly the same for e.g. CIFAR-100. Our surrogate benchmark creation methodology also applies more broadly, beyond image classification tasks. We therefore hope that (1) NAS-Bench-301 will enable the NAS community to evaluate their algorithms on other interesting spaces and not spend their GPU days on the DARTS space by simply using NAS-Bench-301 for benchmarking on it; and that (2) the availability of our methodology for building surrogate NAS benchmarks will form the basis for many interesting benchmarks. Finally, our contributions also encompass the release of all the collected data and the evaluation of numerous performance prediction models on our search space.
> > >
> > > We again thank the reviewer for their feedback and hope that he/she will consider increasing the score after our response.
> > >
> > > -- References --
> > > [1] Yang et al. NAS evaluation is frustratingly hard. In ICLR 2020.
> > > [2] White et al.. Local search is state of the art for nas benchmarks. arXiv preprint 2020.
> > > [3] Zela et al. Understanding and Robustifying Differentiable Architecture Search. In ICLR 2020

---

> ### Author Response · Authors · 2020-11-24
> **Additional questions before the discussion period ends**
>
> We hope that with our response we have addressed the reviewer’s concerns. If there are any additional questions about our paper we would be very happy to answer those before the discussion period ends.

---

### Official Review · AnonReviewer1 · 2020-10-24
**Recommendation to Accept**

**Rating:** 7
**Confidence:** 4

**Review:**

##########################################################################

Summary:

This work filled an important gap in the NAS benchmarks. The previous benchmarks only contain small search space due to the expensive cost of evaluation of neural architecture. In this search space, random search often becomes competitive in the narrow search space. Thus, to provide meaningful comparison, this work provided a benchmark in a large NAS search space (same as in DARTS), and using  surrogate models to predict validation performance of untrained neural architecture. The empirical results suggested using the surrogate benchmarks resulted in similar optimization trajectory as real evaluations and the author also shows one can derive/validate research ideas quickly with the benchmarks.


##########################################################################

Pros:

1. This work filled an important gap in NAS benchmarks.
2. The empirical results are very solid; the observation on the noise in the training is very insightful.
3. The work also contains guidelines for using the benchmarks.
4. The paper is very well written and contains enough detail for reproducibility.

##########################################################################

Cons:

I only have some minor comments:

1. In Section 3.2, first paragraph, the author mentioned that validation and test error are highly correlated. This is clear if the poor performing architectures are included. But in practice, we are only interested in the good performing region, say top 5%, is the correlation still high between validation and test there?

2. In Section 4.2, second paragraph,  I am not sure I understand "We use train/val/test splits (0.8/0.1/0.1) stratified by the NAS
methods used for the data collection".

3. Given the mean and noise estimation based on the surrogate models, is the assumption there is gaussian and every experiment will draw one value from this gaussian? If so, could you state it clearly in the paper? If not, please clarify.

=====POST-REBUTTAL COMMENTS========

Initially I had only minor comments and the authors addressed all of them. I will keep my score.

---

> ### Author Response · Authors · 2020-11-16
> **Response to AnonReviewer1**
>
> We thank the reviewer very much for the positive feedback and for the acceptance score. In the following we address his/her comments and questions.
> 1. **In Section 3.2, first paragraph, the author mentioned that validation and test error are highly correlated. This is clear if the poor performing architectures are included. But in practice, we are only interested in the good performing region, say top 5%, is the correlation still high between validation and test there?**:
> We calculated the correlation between validation and test accuracy on the top 5% of architectures (w.r.t. validation accuracy) as predicted by our surrogate XGBoost model to be 0.37 (Kendall tau) / 0.52 (Spearman rank corr.). As comparison, we also calculated this for NAS-Bench-101 which yielded 0.26 (Kendall Tau) / 0.37 (Spearman rank corr.) and NAS-Bench-201 which yielded a comparable 0.47 (kendall tau) / 0.65 (Spearman rank corr.).
>
> 2. **In Section 4.2, second paragraph, I am not sure I understand "We use train/val/test splits (0.8/0.1/0.1) stratified by the NAS methods used for the data collection".**: We apologize for not being clear enough. This means that in each of the train/val/test splits the fraction of architectures corresponding to a particular optimizer is the same as in the union of train, val and test. For example, we use RS to sample 50% of the total architectures we train and evaluate. Each of the train, val and test will thus contain 50% of the architectures coming from RS. Equivalently, if the train:val:test ratio is 8:1:1, then 80% of the architectures collected from every NAS optimizer will be included in the training set. We added a more thorough description of this point in Section 4.2 of the updated paper.
>
> 3. **Given the mean and noise estimation based on the surrogate models, is the assumption there is gaussian and every experiment will draw one value from this gaussian? If so, could you state it clearly in the paper? If not, please clarify.**: Yes indeed; we had briefly stated this in Section 4.4, but we added a more detailed statement now. The performance is queried from the surrogate ensemble by sampling from the predictive distribution which is modelled as a normal distribution.

---

### Official Review · AnonReviewer4 · 2020-10-29
**Great effort! Some remaining questions.**

**Rating:** 8
**Confidence:** 5

**Review:**

****Update after rebuttal ****
I am increasing my rating for the paper as they did all the experiments I had asked for and updated the paper accordingly.
**************************

Summary:

While NAS has made tremendous advances in recent past, benchmarking algorithms with respect to each other still remains a challenge. Tabular benchmarks like 101 and 201, take a search space and train all possible architectures in them. While this is possible to do for relatively small search spaces and datasets, this is impractical to repeat for larger search spaces (e.g. DARTS' search space which has 10^18 architectures).

This work makes the nice observation that a tabular benchmark treats each architecture as an independent random variable and doesn't utilize any similarities between them. Due to similarities in architecture space, knowing the train/val/test accuracy of one architecture tells us a lot about other architectures nearby. So a predictive model trained on a sparse subset of architectures can actually outperform an exhaustive tabular method.

Lots of careful experiments are reported on the DARTS search space to create predictive models which can accurately predict architecture performance (accuracy, latency) and hence can be used as a 'simulator' by NAS algorithms for rapid research and fair comparison.

Comments:

- The paper is generally well-written and has thorough clean experiments! Thanks!

- My main concern is the following: The fact that the benchmark has added architectures encountered on the trajectory of well-known performing optimizers bothers me a bit. In the ideal world a benchmark should have no knowledge of any particular solution to the problem. This is true in the case of tabular benchmarks like 101 and 201. I understand the position that a surrogate model should be very good at predicting parts of the architecture space which optimizers are most likely to visit. Can we construct surrogates without knowing anything about any particular optimizers the community may invent in the future?

One part of an ablation study answering this question has been presented in Appendix E.2 where a model has been only trained on well-performing architectures (above 92% accuracy) and in Figure 21 has been found adequate for predicting the trajectories of BANANAS and Random Search (RS). This begs the question of what if we only used random sampling to fit surrogate models? (Of course coverage methods like adaptive submodularity-based greedy algorithms may result in even better performance). But can we do the easy baselines first for which the authors already have the data:

1.  Train surrogate model using only the 23746 architectures via random sampling and plot the same figures as in Fig 4 center and right. How much worse are they from current ones?

2. (if compute allows): Use up the entire budget of ~60k architectures in Table 2 only from random sampling and plot Fig 4 center and right.  How much worse are they from current ones?

3. (to have a fair comparison of current method to baseline 1 without using much more compute): Keep the total budget 23746 but fill them up in the same proportion from each method as currently in Table 2. For example RS will have 23746/~60k ratio, DE will have 7275/~60k ratio, etc. This will create "NASBench-301-small" which will have the same budget as baseline 1 above. Then plot Fig 4 center and right with both this and 1.

These might be competitive baselines because the Once-For-All work from Cai et al. ICLR 2020 uses a regressor trained on 16k architectures sampled from a supergraph as a surrogate model to run evolutionary search against and obtain good performance. Also in this paper itself if more than 21500 architectures on 101 are used to train (unclear from the paper whether they were randomly sampled and diverged models rejected or some other technique was used to select them, since it says "subsets of D^{train}" but I think they were randomly sampled, right?), then that itself is better than the tabular benchmark.

Happy to be convinced if these are fair baselines or not. (Also possible that these are already included and I missed them. Ther are a lot of experiments :-))

-My other worry is that the LOOO ablation may be misleading since reasonable optimizers may be visiting similar parts of the architecture space hence may give a false sense of extrapolation.

---

> ### Author Response · Authors · 2020-11-16
> **Response to AnonReviewer4**
>
> We thank the reviewer for reading the paper thoroughly, the positive feedback, and for the detailed and very useful review. We now reply to the questions:
>
> 1. **My main concern is the following: The fact that the benchmark has added architectures encountered on the trajectory of well-known performing optimizers bothers me a bit. In the ideal world a benchmark should have no knowledge of any particular solution to the problem. This is true in the case of tabular benchmarks like 101 and 201. I understand the position that a surrogate model should be very good at predicting parts of the architecture space which optimizers are most likely to visit. Can we construct surrogates without knowing anything about any particular optimizers the community may invent in the future?**: We think this is a very interesting question and we agree that a benchmark ideally should have no prior knowledge about any particular solutions of a NAS optimizer. To this end we conducted the Leave One Optimizer Out (LOOO) experiment in Section 5.1 to remove some of this bias when evaluating the soundness of NAS-Bench-301. It is true that most of the NAS optimizers we use after some time find really competitive architectures, but as one can see from the t-SNE plots in Fig. 9 (Appendix C.4) these optimizers exploit different good-performing regions of the architecture space (e.g., compare DE and BANANAS). Finally, we also think that another approach that might yield better “unbiased” benchmarks could be to use more sophisticated space-filling methods like quasi-random sequences, e.g. Sobol sequences [1], Latin Hypercubes [2] or Adaptive Submodularity [3].
>
> 2. **On the reviewer suggestion to: - Train surrogate model using only the 23746 architectures via random sampling and plot the same figures as in Fig 4 center and right. How much worse are they from current ones? - (to have a fair comparison of current method to baseline 1 without using much more compute): Keep the total budget 23746 but fill them up in the same proportion from each method as currently in Table 2. For example RS will have 23746/60k ratio, DE will have 7275/60k ratio, etc. This will create "NASBench-301-small" which will have the same budget as baseline 1 above. Then plot Fig 4 center and right with both this and 1.**: This is a very interesting experiment. Thank you for the suggestion! We are running this, and we will post a reply when we have these plots ready to update the paper accordingly.
>
> 3. **(if compute allows): Use up the entire budget of ~60k architectures in Table 2 only from random sampling and plot Fig 4 center and right. How much worse are they from current ones?**: Unfortunately, as the reviewer assumed, our compute resources do not allow us to collect this many architectures during this short time frame. However, this is an interesting experiment for the future since anyway we are planning to increase the coverage of the space in the long run.
>
> 4. **Also in this paper itself if more than 21500 architectures on 101 are used to train (unclear from the paper whether they were randomly sampled and diverged models rejected or some other technique was used to select them, since it says "subsets of D^{train}" but I think they were randomly sampled, right?), then that itself is better than the tabular benchmark.**: The reviewer is correct, the architectures in the training subsets were uniformly sampled at random. However, prior to sampling, any architectures in which one of the three evaluations diverged, were removed (as done in e.g. [4]).
>
> 5. **My other worry is that the LOOO ablation may be misleading since reasonable optimizers may be visiting similar parts of the architecture space hence may give a false sense of extrapolation.**: This is a fair point, however it might also be the case that there exist many distinct regions in the architecture space that yield architecture with good and similar performance. We think this is the case for the DARTS search space and the t-SNE plots for the individual optimizers in Figure 9 indicate that certain optimizers (e.g. BANANAS) visit distinct subspaces and are still predicted accurately in the LOOO (Figure 5).
>
> -- References --
>
> [1] M. Sobol. On the distribution of points in a cube and the approximate evaluation of integrals. Zhurnal Vychislitel`noi Matematiki i Matematicheskoi Fiziki. 7(4):784-802, 1967.
> [2] McKay et al. A comparison of three methods for selecting values of input variables in the analysis of output from a computer code. Technometrics, 2000.
> [3] Golovin and Krause. Adaptive submodularity: Theory and applications in active learning and stochastic optimization. Journal of Artificial Intelligence Research, 42:427-486, 2001.
> [4] Wen et al. Neural predictor for neural architecture search. In ECCV 2020.

---

> > ### Author Response · Authors · 2020-11-23
> > **Reply to AnonReviewer4 to update on suggested experiments**
> >
> > We would like to thank the reviewer again for the constructive feedback and suggesting very insightful experiments. We have performed the experiments 1 and 3  suggested by the reviewer and have updated the paper accordingly (in particular we added Appendix E.3 and updated our guidelines for creating surrogate NAS benchmarks).
> >
> > In summary, we found that a surrogate NAS benchmark trained only on architectures obtained via random sampling does indeed suffice to yield plots for comparing NAS optimizers over time that look very similar to the real benchmark (Figure 24 in Appendix E.3), similarly to what we observed when only training on the well-performing part of the space (Figure 22 in Appendix E.2). However, as expected, the underlying surrogate models become more accurate when including the additional architectures found by the other NAS optimizers (Figure 23 in Appendix E.3).
> >
> > We agree with the reviewer’s argument that NAS benchmarks that rely on training architectures generated by NAS optimizers may yield more reliable predictions for the type of architectures that tend to be visited by these optimizers, and that it would therefore be desirable to construct NAS benchmarks free of such potential bias. Having seen (in the experiment above) that such NAS benchmarks actually *can* yield surprisingly realistic comparisons of NAS optimizers, we now also release versions of our benchmarks based on only this random training data. We also updated our guidelines for the creation of NAS benchmarks (Appendix F) as follows:
> > “Also, in order to guard against a possible case of “bias” in a surrogate benchmark (in the sense of making more accurate predictions for architectures explored by a particular type of NAS optimizer), we recommend to provide two versions of a surrogate: one based on all available training architectures (including those found by NAS optimizers), and one based only on the data gathered for overall coverage (1. above).”
> >
> > We would be very interested in hearing whether the reviewer agrees with our conclusions and in feedback on this adjustment to our guidelines. Thanks again for suggesting this insightful experiment!

---

> > > ### Comment · AnonReviewer4 · 2020-11-23
> > > **Great execution!**
> > >
> > > First of all thanks for all the hard work! I know turning these experiments around in such short time frame is not easy.
> > >
> > > Secondly, in my opinion the case for surrogate models as benchmarks has only been made stronger as a result of these experiments. The fact that we can create surrogate models which do not have to have carefully thought out samples from solutions to the NAS problem makes it even more attractive. I think the authors should *proudly* make this case strongly in the main body of the paper itself and not just in the appendix. Especially in section 7 on guidelines.
> > >
> > > I will update my score for this paper accordingly! :-)

---

> > > > ### Author Response · Authors · 2020-11-23
> > > > **Thank you for the suggestion!**
> > > >
> > > > We would like to thank the reviewer very much for the quick turn-around and for appreciating our work! As the reviewer suggested we have now added the following new bullet point in the best practice list of Section 7:
> > > >
> > > > *“We encourage running experiments on versions of NAS-Bench-301 (and other, future NAS surrogate benchmarks) that are based on (1) all available training architectures and (2) only architectures collected with uninformed methods, such as random search or space-filling designs. As shown in Appendix E.3, (1) yields better predictive models, but (2) avoids any potential bias (in the sense of making more accurate predictions for architectures explored by a particular type of NAS optimizer) and can still yield strong benchmarks.”*
> > > >
> > > > We had already added the following paragraph at the end of Section 5.1 in our revision (apologies for not mentioning this in the previous reply):
> > > >
> > > > *“We also investigate whether it is possible to create benchmarks only on random architectures in Appendix E.3, and find that we can indeed obtain realistic trajectories but lose some predictive performance in the well-performing regions. Nevertheless, such benchmarks have the advantage of not possibly favouring any NAS optimizer used for the generation of training data, and we thus recommend their release in addition to the benchmarks based on the full training data.”*,
> > > >
> > > > but mentioning it again in Section 7 is a very good suggestion to provide a concrete guideline.
> > > > Another change that we did not mention in our previous reply is that we had extended the last bullet point in the best practice list of Section 7 with 2 new released versions of NB301, namely the ones with surrogates trained only with the randomly sampled architectures.

---

> > > > > ### Comment · AnonReviewer4 · 2020-11-23
> > > > > **One last thing!**
> > > > >
> > > > > Thanks for the changes to the paper.
> > > > >
> > > > > Is it possible to release the training logs as well for the set of sampled architectures in the benchmark? (This might already be there in the benchmark). The training logs (especially error vs. epoch) can be very useful as well in NAS research independent of the benchmark.

---

> > > > > > ### Author Response · Authors · 2020-11-24
> > > > > > **Releasing nasbench301_full_data**
> > > > > >
> > > > > > Thank you for the suggestion! We were indeed planning to release the training logs for the community, which include learning curves for train, validation and test sets as well as other statistics such as number of model parameters, runtime, etc. We are currently compiling the dataset (this takes a couple of hours) which will be uploaded at this figshare account: https://figshare.com/authors/anon_ymous/9470543, with the name *nasbench301_full_data* and it will be available before the author discussion period ends today (November 24th anywhere on Earth).

---

### Official Review · AnonReviewer3 · 2020-11-02
**The benefits of introducing a new benchmark could be better established**

**Rating:** 5
**Confidence:** 4

**Review:**


Summary:

The authors propose a new benchmark for evaluating surrogate functions for architecture search. According to the authors, existing tabular architecture search benchmarks are insufficient for this purpose due to using overly small search spaces. The main difference of this benchmark and other existing architecture search benchmarks such as NAS Bench 101 and NAS Bench 201 is that they do not attempt to evaluate all the architectures in the search space and do so for a much larger search space (DARTS). The authors then use this new dataset to show that surrogate functions are better than tabular estimators (error wise; although some lower performance architectures were discarded to make this case). Additionally, the authors compare the proposed benchmark (based on surrogate functions) with a real benchmark and observe that the results are qualitatively similar. Finally, the authors show that reevaluate the claim that local search is state-of-the-art for architecture search and find that, using their benchmark, that local search is still state-of-the-art provided that enough computational budget is available for the experiment.

Pros:
+ The paper presents a new dataset for architecture search that does not rely on exhaustive evaluation of all architectures in the search space. The experiments conducted that compare different surrogate functions on this benchmark are solid
+ The paper suggests that building benchmarks for different search spaces can be accomplished through the surrogate function route where first a dataset of architectures is collected and then it is used to train a surrogate model. The surrogate function is then used as an interface between the search space and the search algorithm. This approach for building benchmarks is general and is likely to be useful in the construction of future benchmarks for architecture search.

Cons:
- The fact that existing benchmarks for architecture search are insufficient for surrogate function evaluation and lead to wrong inferences is perhaps insufficiently supported. It is well known in architecture search work that existing search spaces (DARTS being one of them) have limited performance variability and that much of the performance variation ascribed to different architecture search methods can often be ascribed to differences in search spaces. Is a single search space good enough to mitigate these problems? For example, how do we guarantee that NAS Bench 301 is not just another dataset and guarantee that addresses some of the perceived problems with existing architecture search search spaces and benchmarks?


Comments:
- Appendices weren't included in the submission.
- It would be interesting to discuss how this benchmark should be placed in the context of other existing benchmarks for architecture search to guide future practice.
- The motivation from Section 2 is rather sparse. It would be warranted to show that this trend is consistent with other metrics such as squared error and Kendall Tau (i.e., showing the ranks of different architectures are also preserved better).
- Insufficient information about the feature representation that is used for prediction?

---

> ### Author Response · Authors · 2020-11-16
> **Response to AnonReviewer3 (Part 1/3)**
>
> We would like to thank the reviewer for taking his/her time to review our paper and for the useful feedback and suggestions. We hope that the reviewer will consider increasing his/her score after our response to his concerns which we provide in the following.
>
> 1. **The fact that existing benchmarks for architecture search are insufficient for surrogate function evaluation and lead to wrong inferences is perhaps insufficiently supported.**: The main difference between existing benchmarks and realistic search spaces is their size mismatch. This in turn leads to 1. NAS algorithms like local search to become very inefficient [1] and 2. conclusions and hypotheses drawn from these benchmarks potentially being distinct. We do not argue that existing tabular benchmarks are not interesting (maybe easier) spaces to benchmark surrogate models and we have already seen papers applying GNNs to learn useful representations on these tabular benchmarks [4, 5]. What we argue is that surrogate models can effectively represent the NAS search space of these tabular benchmarks by using just a fraction of the architectures in search space (Figure 1 in the paper) and provide a better estimate for the test performance compared to the tabular entries (Table 1). We show that regression models can effectively model the noisy evaluations stemming from the SGD training of the neural networks better than a table which contains only a limited number of point estimates.
>
> 2. **It is well known in architecture search work that existing search spaces (DARTS being one of them) have limited performance variability**: We agree that many existing search spaces (including the one of DARTS) do not yield the largest variety in performances. Before we invested the compute to train the architectures in the DARTS space, we thought hard about whether we should introduce a new search space and then create a surrogate of that, but that would come with all kinds of confounding factors, and we ultimately decided that it is far more useful (and convincing) to demonstrate the concept of surrogate NAS benchmarks on a highly used benchmark. This is also because DARTS is still the most widely used non-tabular search space and many current NAS papers are burning very substantial GPU time on this benchmark.
> We would also like to note that the performance depends on the macro architecture and the choice of training pipeline. Figure 8 in the appendix shows that our search space yields architectures with performances between 80-95 % accuracy. Note our training dataset does not encompass all architectures in the space. Since we focus on good regions the bad performing architectures are underrepresented.

---

> > ### Author Response · Authors · 2020-11-16
> > **Response to AnonReviewer3 (Part 2/3)**
> >
> > 3. **Much of the performance variation ascribed to different architecture search methods can often be ascribed to differences in search spaces. Is a single search space good enough to mitigate these problems? For example, how do we guarantee that NAS Bench 301 is not just another dataset and guarantee that addresses some of the perceived problems with existing architecture search search spaces and benchmarks?**:
> > Thank you for raising this point! We agree with the reviewer that having a variety of benchmarks is very beneficial for thorough comparison of different methods. This is also the case for other domains, with e.g. OpenML benchmarking suites for tabular classification [2] or HPOlib for hyperparameter optimization [3]. We agree that the NAS research community *must* move forward to evaluate methods on a variety of benchmarks, and the way to actually make this feasible (without unreasonable computational requirements and carbon emissions) is through fast-to-evaluate benchmarks in the form of tabular and surrogate benchmarks. While only using tabular benchmarks would tie the community to small search spaces that are feasible to evaluate exhaustively, with surrogate benchmarks we’re opening up the opportunity to also create fast-to-evaluate benchmarks with large search spaces. Therefore, we see the creation of NAS-Bench-301 firmly on the critical path towards the broad variety of fast-to-evaluate benchmarks the community needs. The availability of such a broad set of fast-to-evaluate benchmarks will offer researchers a cheap way to prototype and benchmark on this space and allow them to invest the compute that they do have into tackle additional challenging and interesting applications and search spaces.
> > Finally, we fully agree with the first part of this comment that much of the performance variation in published performance numbers (e.g., on CIFAR-10) ascribed to different architecture search methods can often be ascribed to differences in search spaces instead. However, this is for comparing the results achieved by method A run on search space A in paper A to the results achieved by method B on search space B in paper B. We certainly do not propose that researchers use NAS-Bench-301 in their papers and then compare to the results other authors obtained on other benchmarks. Rather, we propose that NAS-Bench-301, like the tabular benchmarks NAS-Bench-101 and NAS-Bench-201 becomes part of the typical benchmarks to be run, as this can be done largely for free, and that results on NAS-Bench-301 in one paper are compared to results on NAS-Bench-301 in another paper. The more fast-to-evaluate benchmarks we create the more comprehensive this set of basically free benchmarks becomes, and the more reliable the empirical findings of the community become.
> >
> > 4. **Appendices weren't included in the submission**: We’re sorry for the confusion; we did in fact include the appendix in the original submission, following the second option given for this in the author guidelines (https://iclr.cc/Conferences/2021/AuthorGuide) to submit them as part of the zip file for supplementary material. While we chose this option since this is also the standard process for other conferences, we fully agree that that is vastly less convenient than simply attaching it to the main paper, and we now did that.
> >
> > 5. **It would be interesting to discuss how this benchmark should be placed in the context of other existing benchmarks for architecture search to guide future practice.**: Thank you for the remark, we have updated the conclusion to highlight that our benchmark can be used for cheap, fast runs for developing and testing new black-box NAS algorithms or for monitoring one-shot optimizers during the search phase which very often lead to degenerated architectures [6] (which is why it is of high interest to evaluate the anytime performance of such methods). We also mentioned the prospect of the community developing a broad, varied, set of fast-to-evaluate benchmarks using different types of applications and search spaces, and that surrogate NAS benchmarks allow including large search spaces in this set.
> >
> > 6. **The motivation from Section 2 is rather sparse. It would be warranted to show that this trend is consistent with other metrics such as squared error and Kendall Tau (i.e., showing the ranks of different architectures are also preserved better).**: Thank you for the suggestion. We evaluated the experiment with the MSE and Kendall Tau as you suggested and added the results in Table 6 in the Appendix. The surrogate model yields better performance in all cases.
> >
> > 7. **Insufficient information about the feature representation that is used for prediction?**: We extended the description of the feature representations used for the Graph Neural Network and the other surrogate models in Appendix D.1 with more details on the feature representations. Our apologies again that the appendix was “hidden” in the zip file before.

---

> > > ### Author Response · Authors · 2020-11-16
> > > **Response to AnonReviewer3 (Part 3/3)**
> > >
> > > -- References --
> > > [1] White et al.. Local search is state of the art for nas benchmarks. arXiv preprint 2020.
> > > [2] Bischl et al. OpenML Benchmarking Suites, arXiv preprint arXiv:1708.03731, 2019.
> > > [3] Eggensperger et al. Towards an Empirical Foundation for Assessing Bayesian Optimization of Hyperparameters. NeurIPS BayesOpt Workshop. 2013
> > > [4] Zhang et al. D-VAE: A Variational Autoencoder for Directed Acyclic Graphs. In NeurIPS 2019
> > > [5] Yan et al. Does Unsupervised Architecture RepresentationLearning Help Neural Architecture Search? In NeurIPS 2020
> > > [6] Zela et al. Understanding and Robustifying Differentiable Architecture Search. In ICLR 2020

---

> ### Author Response · Authors · 2020-11-24
> **Additional questions before discussion period ends**
>
> If the reviewer has any additional questions about our paper we would be very happy to answer those before the discussion period ends.

---

### Author Response · Authors · 2020-11-23
**Summary of changes to the paper**

We would like to thank the reviewers again for their feedback and important suggestions. Below we highlight the main changes made to our paper:
1. We have added Table 6 in the Appendix that shows additional metrics for the experiment from the motivation section. *(suggested from AnonReviewer3)*
2. We have added a new section (Appendix E.3) with experiments with surrogates fitted only on RS data / on a small NB301 version. *(suggested from AnonReviewer4)*
3. We have added more details on the representations used for the surrogates in Appendix D.1. *(suggested from AnonReviewer3)*
4. We have provided more clarifications within the text and made the figures larger.
5. We have extended further Appendix F with the guidelines on how to construct surrogate benchmarks.

If the reviewers have any further additional questions, generally about the latest version of our paper or our individual responses, we are happy to answer those before the discussion period ends.

---

### Decision · Program_Chairs · 2021-01-07
**Final Decision**

**Decision:**

Reject

**Comment:**

The contributions of this paper lie in two areas: a new benchmarking dataset and a new way to generate benchmarking datasets. Overall, the reviewers are split in their assessment based on which particular area they are focusing on. Reviewers who focus more on evaluating this work as a new benchmarking dataset, correctly point out that the variation within the search space has been shown to be limited and that the evaluation focuses on an overly-studied and toy (by today’s standards) dataset such a CIFAR-10. In terms of choice of dataset, this work is indeed a step backwards from nasbench201, which includes more datasets, although it is a step forward in terms of size of the search space. As one reviewer correctly points out “This paper doesn’t present a benchmark. It provides a model that represents computationally-efficient means of getting network accuracies from the DARTS search space”. The authors argue that the combination of the DARTS search space and its evaluation on CIFAR-10 is the de-facto evaluation standard in the NAS community, which is also true. Ultimately, benchmark datasets somewhat direct the attention of the community and this attention would be better directed elsewhere, not on DARTS+CIFAR-10, as pointed out by some of the reviewers.

On the other hand, this work is as much about a new benchmark as it is about a *protocol* to generate new benchmarks. Specifically, a big part of the appeal and novelty here lie in the idea of training a predictive model from a small subset of architecture evaluations. From this perspective, the authors showed evidence that their approach is sound, economical (in terms of computational cost) and robust to sources of bias in the selection of the architectures to evaluate. The limitation here is that this was only shown in search spaces that where either small or lacking diversity and thus it’s unclear how general its findings are.

Overall, this is very much a paper that could have gone either way in terms of acceptance. It’s a step in the right direction in terms of methodology that can be used to generate reproducible benchmarks in a computationally efficient way. It’s a step backwards in terms renewing focus on measures of performance that (arguably) we have all overfit to.